# A Unified Convergence Theorem for Stochastic Optimization Methods

**Xiao Li**
School of Data Science (SDS)
Shenzhen Institute of Artificial Intelligence
and Robotics for Society (AIRS)
The Chinese University of Hong Kong, Shenzhen
Shenzhen, China
`lixiao@cuhk.edu.cn`

**Andre Milzarek**
School of Data Science (SDS)
Shenzhen Research Institute of Big Data (SRIBD)
The Chinese University of Hong Kong, Shenzhen
Shenzhen, China
`andremilzarek@cuhk.edu.cn`

## Abstract

In this work, we provide a fundamental unified convergence theorem used for deriving expected and almost sure convergence results for a series of stochastic optimization methods. Our unified theorem only requires to verify several representative conditions and is not tailored to any specific algorithm. As a direct application, we recover expected and almost sure convergence results of the stochastic gradient method (SGD) and random reshuffling (RR) under more general settings. Moreover, we establish new expected and almost sure convergence results for the stochastic proximal gradient method (prox-SGD) and stochastic model-based methods for nonsmooth nonconvex optimization problems. These applications reveal that our unified theorem provides a plugin-type convergence analysis and strong convergence guarantees for a wide class of stochastic optimization methods.

## 1 Introduction

Stochastic optimization methods are widely used to solve stochastic optimization problems and empirical risk minimization, serving as one of the foundations of machine learning. Among the many different stochastic methods, the most classic one is the stochastic gradient method (SGD), which dates back to Robbins and Monro [36]. If the problem at hand has a finite-sum structure, then another popular stochastic method is random reshuffling (RR) [20]. When the objective function has a composite form or is weakly convex (nonsmooth and nonconvex), then the stochastic proximal gradient method (prox-SGD) and stochastic model-based algorithms are the most typical approaches [18, 11]. Apart from the mentioned stochastic methods, there are many others like SGD with momentum, Adam, stochastic higher order methods, etc. In this work, our goal is to establish and understand fundamental *convergence* properties of these stochastic optimization methods via a novel unified convergence framework.

**Motivations.** Suppose we apply SGD to minimize a smooth nonconvex function $f$. SGD generates a sequence of iterates $\{x^k\}_{k\geq 0}$, which is a stochastic process due to the randomness of the algorithm and the utilized stochastic oracles. The most commonly seen 'convergence result' for SGD is the

36th Conference on Neural Information Processing Systems (NeurIPS 2022).

*expected iteration complexity*, which typically takes the form [17]

$$\min_{k=0,\ldots,T} \; \mathbb{E}[\|\nabla f(\boldsymbol{x}^k)\|^2] \leq \mathcal{O}\left(\frac{1}{\sqrt{T+1}}\right) \quad \text{or} \quad \mathbb{E}[\|\nabla f(\boldsymbol{x}^{\bar{k}})\|^2] \leq \mathcal{O}\left(\frac{1}{\sqrt{T+1}}\right), \quad (1)$$

where $T$ denotes the total number of iterations and $\bar{k}$ is an index sampled uniformly at random from $\{0,\ldots,T\}$. Note that we ignored some higher-order convergence terms and constants to ease the presentation. Complexity results are integral to understand core properties and progress of the algorithm during the first $T$ iterations, while the asymptotic convergence behavior plays an equally important role as it characterizes whether an algorithm can eventually approach an exact stationary point or not. We refer to Appendix H for additional motivational background for studying asymptotic convergence properties of stochastic optimization methods. Here, an *expected convergence result*, associated with the nonconvex minimization problem $\min_{\boldsymbol{x}} f(\boldsymbol{x})$, has the form

$$\lim_{k\to\infty} \mathbb{E}[\|\nabla f(\boldsymbol{x}^k)\|] = 0. \quad (2)$$

Intuitively, it should be possible to derive expected convergence from the expected iteration complexity (1) by letting $T \to \infty$. However, this is not the case as the 'min' operator and the sampled $\bar{k}$ are not well defined or become meaningless when $T$ goes to $\infty$.

The above results are stated in expectation and describe the behavior of the algorithm by averaging infinitely many runs. Though this is an important convergence measure, in practical situations the algorithm is often only run once and the last iterate is returned as a solution. This observation motivates and necessitates *almost sure convergence results*, which establish convergence with probability 1 for a single run of the stochastic method:

$$\lim_{k\to\infty} \|\nabla f(\boldsymbol{x}^k)\| = 0 \quad \text{almost surely}. \quad (3)$$

**Backgrounds.** Expected and almost sure convergence results have been extensively studied for convex optimization; see, e.g., [10, 34, 42, 46, 5, 41]. Almost sure convergence of SGD for minimizing a smooth nonconvex function $f$ was provided in the seminal work [3] using very standard assumptions, i.e., Lipschitz continuous $\nabla f$ and bounded variance. Under the same conditions, the same almost sure convergence of SGD was established in [33] based on a much simpler argument than that of [3]. A weaker 'lim inf'-type almost sure convergence result for SGD with AdaGrad step sizes was shown in [26]. Recently, the work [28] derives almost sure convergence of SGD under the assumptions that $f$ and $\nabla f$ are Lipschitz continuous, $f$ is coercive, $f$ is not asymptotically flat, and the $\upsilon$-th moment of the stochastic error is bounded with $\upsilon \geq 2$. This result relies on stronger assumptions than the base results in [3]. Nonetheless, it allows more aggressive diminishing step sizes if $\upsilon > 2$. Apart from standard SGD, almost sure convergence of different respective variants for min-max problems was discussed in [22]. In terms of expected convergence, the work [6] showed $\lim_{k\to\infty} \mathbb{E}[\|\nabla f(\boldsymbol{x}^k)\|] = 0$ under the additional assumptions that $f$ is twice continuously differentiable and the multiplication of the Hessian and gradient $\nabla^2 f(\boldsymbol{x})\nabla f(\boldsymbol{x})$ is Lipschitz continuous.

Though the convergence of SGD is well-understood and a classical topic, asymptotic convergence results of the type (2) and (3) often require a careful and separate analysis for other stochastic optimization methods — especially when the objective function is simultaneously nonsmooth and nonconvex. In fact and as outlined, a direct transition from the more common complexity results (1) to the full convergence results (2) and (3) is often not possible without further investigation.

**Main contributions.** We provide a fundamental *unified convergence theorem* (see Theorem 2.1) for deriving both expected and almost sure convergence of stochastic optimization methods. Our theorem is not tailored to any specific algorithm, instead it incorporates several abstract conditions that suit a vast and general class of problem structures and algorithms. The proof of this theorem is elementary.

We then apply our novel theoretical framework to several classical stochastic optimization methods to recover existing and to establish new convergence results. Specifically, we recover expected and almost sure convergence results for SGD and RR. Though these results are largely known in the literature, we derive unified and slightly stronger results under a general ABC condition [24, 23] rather than the standard bounded variance assumption. We also remove the stringent assumption used in [6] to show (2) for SGD. As a core application of our framework, we derive expected and almost sure convergence results for prox-SGD in the nonconvex setting and under the more general ABC condition and for stochastic model-based methods under very standard assumptions. In particular, we show that the iterates $\{\boldsymbol{x}^k\}_{k\geq 0}$ generated by prox-SGD and other stochastic model-based methods

will approach the set of stationary points almost surely and in an expectation sense. These results are *new* to our knowledge (see also Subsection 3.5 for further discussion).

The above applications illustrate the general plugin-type purpose of our unified convergence analysis framework. Based on the given recursion and certain properties of the algorithmic update, we can derive broad convergence results by utilizing our theorem, which can significantly simplify the convergence analysis of stochastic optimization methods; see Subsection 2.1 for a summary.

## 2 A unified convergence theorem

Throughout this work, let $(\Omega, \mathcal{F}, \{\mathcal{F}_k\}_{k\geq 0}, \mathbb{P})$ be a filtered probability space and let us assume that the sequence of iterates $\{\boldsymbol{x}^k\}_{k\geq 0}$ is adapted to the filtration $\{\mathcal{F}_k\}_{k\geq 0}$, i.e., each of the random vectors $\boldsymbol{x}^k : \Omega \to \mathbb{R}^n$ is $\mathcal{F}_k$-measurable.

In this section, we present a unified convergence theorem for the sequence $\{\boldsymbol{x}^k\}_{k\geq 0}$ based on an abstract convergence measure $\boldsymbol{\Phi}$. To make the abstract convergence theorem more accessible, the readers may momentarily regard $\boldsymbol{\Phi}$ and $\{\mu_k\}_{k\geq 0}$ as $\nabla f$ and the sequence related to the step sizes, respectively. We then present the main steps for showing the convergence of a stochastic optimization method by following a step-by-step verification of the conditions in our unified convergence theorem.

**Theorem 2.1.** *Let the mapping* $\boldsymbol{\Phi} : \mathbb{R}^n \to \mathbb{R}^m$ *and the sequences* $\{\boldsymbol{x}^k\}_{k\geq 0} \subseteq \mathbb{R}^n$ *and* $\{\mu_k\}_{k\geq 0} \subseteq \mathbb{R}_{++}$ *be given. Consider the following conditions:*

(P.1) *The function* $\boldsymbol{\Phi}$ *is* $\mathsf{L}_\Phi$-*Lipschitz continuous for some* $\mathsf{L}_\Phi > 0$, *i.e., we have* $\|\boldsymbol{\Phi}(\boldsymbol{x}) - \boldsymbol{\Phi}(\boldsymbol{y})\| \leq \mathsf{L}_\Phi \|\boldsymbol{x} - \boldsymbol{y}\|$ *for all* $\boldsymbol{x}, \boldsymbol{y} \in \mathbb{R}^n$.

(P.2) *There exists a constant* $a > 0$ *such that* $\sum_{k=0}^{\infty} \mu_k \mathbb{E}[\|\boldsymbol{\Phi}(\boldsymbol{x}^k)\|^a] < \infty$.

*The following statements are valid:*

(i) *Let the conditions* (P.1)–(P.2) *be satisfied and suppose further that*

(P.3) *There exist constants* $\mathsf{A}, \mathsf{B}, b \geq 0$ *and* $p_1, p_2, q > 0$ *such that*
$$\mathbb{E}[\|\boldsymbol{x}^{k+1} - \boldsymbol{x}^k\|^q] \leq \mathsf{A}\mu_k^{p_1} + \mathsf{B}\mu_k^{p_2} \mathbb{E}[\|\boldsymbol{\Phi}(\boldsymbol{x}^k)\|^b].$$

(P.4) *The sequence* $\{\mu_k\}_{k\geq 0}$ *and the parameters* $a, b, q, p_1, p_2$ *satisfy*
$$\{\mu_k\}_{k\geq 0} \text{ is bounded}, \quad \sum_{k=0}^{\infty} \mu_k = \infty, \quad \text{and} \quad a, q \geq 1, \ a \geq b, \ p_1, p_2 \geq q.$$

*Then, it holds that* $\lim_{k\to\infty} \mathbb{E}[\|\boldsymbol{\Phi}(\boldsymbol{x}^k)\|] = 0$.

(ii) *Let the properties* (P.1)–(P.2) *hold and assume further that*

(P.3′) *There exist constants* $\mathsf{A}, b \geq 0$, $p_1, p_2, q > 0$ *and random vectors* $\boldsymbol{A}_k, \boldsymbol{B}_k : \Omega \to \mathbb{R}^n$ *such that*
$$\boldsymbol{x}^{k+1} = \boldsymbol{x}^k + \mu_k^{p_1} \boldsymbol{A}_k + \mu_k^{p_2} \boldsymbol{B}_k$$
*and for all* $k$, $\boldsymbol{A}_k, \boldsymbol{B}_k$ *are* $\mathcal{F}_{k+1}$-*measurable and we have* $\mathbb{E}[\boldsymbol{A}_k \mid \mathcal{F}_k] = 0$ *almost surely,* $\mathbb{E}[\|\boldsymbol{A}_k\|^q] \leq \mathsf{A}$, *and* $\limsup_{k\to\infty} \|\boldsymbol{B}_k\|^q / (1 + \|\boldsymbol{\Phi}(\boldsymbol{x}^k)\|^b) < \infty$ *almost surely.*

(P.4′) *The sequence* $\{\mu_k\}_{k\geq 0}$ *and the parameters* $a, b, q, p_1, p_2$ *satisfy* $\mu_k \to 0$,
$$\sum_{k=0}^{\infty} \mu_k = \infty, \quad \sum_{k=0}^{\infty} \mu_k^{2p_1} < \infty, \quad \text{and} \quad q \geq 2, \ qa \geq b, \ p_1 > \frac{1}{2}, \ p_2 \geq 1.$$

*Then, it holds that* $\lim_{k\to\infty} \|\boldsymbol{\Phi}(\boldsymbol{x}^k)\| = 0$ *almost surely.*

The proof of Theorem 2.1 is elementary. We provide the core ideas here and defer its proof to Appendix A. Item (i) is proved by contradiction. An easy first result is $\liminf_{k\to\infty} \mathbb{E}[\|\boldsymbol{\Phi}(\boldsymbol{x}^k)\|^a] = 0$. We proceed and assume that $\{\mathbb{E}[\|\boldsymbol{\Phi}(\boldsymbol{x}^k)\|]\}_{k\geq 0}$ does not converge to zero. Then, for some $\delta > 0$, we can construct two subsequences $\{\ell_t\}_{t\geq 0}$ and $\{u_t\}_{t\geq 0}$ such that $\ell_t < u_t$ and $\mathbb{E}[\|\boldsymbol{\Phi}(\boldsymbol{x}^{\ell_t})\|] \geq 2\delta$, $\mathbb{E}[\|\boldsymbol{\Phi}(\boldsymbol{x}^{u_t})\|^a] \leq \delta^a$, and $\mathbb{E}[\|\boldsymbol{\Phi}(\boldsymbol{x}^k)\|^a] > \delta^a$ for all $\ell_t < k < u_t$. Based on this construction, the conditions in the theorem, and a set of inequalities, we will eventually reach a contradiction. We notice that the Lipschitz continuity of $\boldsymbol{\Phi}$ plays a prominent role when establishing this contradiction. Our overall proof strategy is inspired by the analysis of classical trust region-type methods, see, e.g., [9, Theorem 6.4.6]. Let us also mention that a different strategy for the fully deterministic setting and

scalar case $\mathbf{\Phi} : \mathbb{R}^n \to \mathbb{R}$ was provided in [8]. For item (ii), we first control the stochastic behavior of the error terms $\mathbf{A}_k$ by martingale convergence theory. We can then conduct sample-based arguments to derive the final result, which is essentially deterministic and hence, follows similar arguments to that of item (i).

The major application areas of our unified convergence framework comprise stochastic optimization methods that have non-vanishing stochastic errors or that utilize diminishing step sizes. In the next subsection, we state the main steps for showing convergence of stochastic optimization methods. This also clarifies the abstract conditions listed in the theorem.

## 2.1 The steps for showing convergence of stochastic optimization methods

In order to apply the unified convergence theorem, we have to verify the conditions stated in the theorem, resulting in three main phases below.

**Phase I: Verifying (P.1)–(P.2).** Conditions (P.1)–(P.2) are used for both the expected and the almost sure convergence results. Condition (P.1) is a problem property and is very standard. We present the final convergence results in terms of the abstract measure $\mathbf{\Phi}$. This measure can be regarded as $f - f^*$ in convex optimization, $\nabla f$ in smooth nonconvex optimization, the gradient of the Moreau envelope in weakly convex optimization, etc. In all the situations, assuming Lipschitz continuity of the convergence measure $\mathbf{\Phi}$ is standard and is arguably a minimal assumption in order to obtain iteration complexity and/or convergence results.

Condition (P.2) is typically a result of the algorithmic property or complexity analysis. To verify this condition, one first establishes the recursion of the stochastic method, which almost always has the form

$$\mathbb{E}[\boldsymbol{y}_{k+1} \mid \mathcal{F}_k] \leq (1 + \beta_k)\boldsymbol{y}_k - \mu_k \|\mathbf{\Phi}(\boldsymbol{x}^k)\|^a + \zeta_k.$$

Here, $\boldsymbol{y}_k$ is a suitable Lyapunov function measuring the (approximate) descent property of the stochastic method, $\zeta_k$ represents the error term satisfying $\sum_{k=0}^{\infty} \zeta_k < \infty$, $\beta_k$ is often related to the step sizes and satisfies $\sum_{k=0}^{\infty} \beta_k < \infty$. Then, applying the supermartingale convergence theorem (see Theorem B.1), we obtain $\sum_{k=0}^{\infty} \mu_k \mathbb{E}[\|\mathbf{\Phi}(\boldsymbol{x}^k)\|^a] < \infty$, i.e., condition (P.2).

Since condition (P.2) is typically a consequence of the underlying algorithmic recursion, one can also derive the standard finite-time complexity bound (1) in terms of the measure $\mathbb{E}[\|\mathbf{\Phi}(\boldsymbol{x}^k)\|^a]$ based on it. Hence, non-asymptotic complexity results are also included implicitly in our framework as a special case. To be more specific, (P.2) implies $\sum_{k=0}^{T} \mu_k \mathbb{E}[\|\mathbf{\Phi}(\boldsymbol{x}^k)\|^a] \leq M$ for some constant $M > 0$ and some total number of iterations $T$. This then yields $\min_{0 \leq k \leq T} \mathbb{E}[\|\mathbf{\Phi}(\boldsymbol{x}^k)\|^a] \leq M / \sum_{k=0}^{T} \mu_k$. Note that the sequence $\{\mu_k\}_{k \geq 0}$ is often related to the step sizes. Thus, choosing the step sizes properly results in the standard finite-time complexity result.

**Phase II: Verifying (P.3)–(P.4) for showing expected convergence.** Condition (P.3) requires an upper bound on the step length of the update in terms of expectation, including upper bounds for the search direction and the stochastic error of the algorithm. It is often related to certain bounded variance-type assumptions for analyzing stochastic methods. For instance, (P.3) is satisfied under the standard bounded variance assumption for SGD, the more general ABC assumption for SGD, the bounded stochastic subgradients assumption, etc. Condition (P.4) is a standard diminishing step sizes condition used in stochastic optimization.

Then, one can apply item (i) of Theorem 2.1 to obtain $\mathbb{E}[\|\mathbf{\Phi}(\boldsymbol{x}^k)\|] \to 0$.

**Phase III: Verifying (P.3′)–(P.4′) for showing almost sure convergence.** Condition (P.3′) is parallel to (P.3). It decomposes the update into a martingale term $\mathbf{A}_k$ and a bounded error term $\mathbf{B}_k$. We will see later that this condition holds true for many stochastic methods. Though this condition requires the update to have a certain decomposable form, it indeed can be verified by bounding the step length of the update in conditional expectation, which is similar to (P.3). Hence, (P.3′) can be interpreted as a conditional version of (P.3). To see this, we can construct

$$\boldsymbol{x}^{k+1} = \boldsymbol{x}^k + \mu_k \cdot \underbrace{\tfrac{1}{\mu_k}\left(\boldsymbol{x}^{k+1} - \boldsymbol{x}^k - \mathbb{E}[\boldsymbol{x}^{k+1} - \boldsymbol{x}^k \mid \mathcal{F}_k]\right)}_{\boldsymbol{A}_k} + \mu_k \cdot \underbrace{\tfrac{1}{\mu_k}\mathbb{E}[\boldsymbol{x}^{k+1} - \boldsymbol{x}^k \mid \mathcal{F}_k]}_{\boldsymbol{B}_k}. \quad (4)$$

By Jensen's inequality, we then have $\mathbb{E}[\boldsymbol{A}_k \mid \mathcal{F}_k] = 0$,

$$\mathbb{E}[\|\boldsymbol{A}_k\|^q] \leq 2^q \mu_k^{-q} \cdot \mathbb{E}[\|\boldsymbol{x}^{k+1} - \boldsymbol{x}^k\|^q], \quad \text{and} \quad \|\boldsymbol{B}_k\|^q \leq \mu_k^{-q} \cdot \mathbb{E}[\|\boldsymbol{x}^{k+1} - \boldsymbol{x}^k\|^q \mid \mathcal{F}_k].$$

Thus, once it is possible to derive $\mathbb{E}[\|\boldsymbol{x}^{k+1} - \boldsymbol{x}^k\|^q \mid \mathcal{F}_k] = \mathcal{O}(\mu_k^q)$ in an almost sure sense, condition (P.3′) is verified with $p_1 = p_2 = 1$. Condition (P.4′) is parallel to (P.4) and is standard in stochastic optimization. Application of item (ii) of Theorem 2.1 then yields $\|\boldsymbol{\Phi}(\boldsymbol{x}^k)\| \to 0$ almost surely.

In the next section, we will illustrate how to show convergence for a set of classic stochastic methods by following the above three steps.

## 3    Applications to stochastic optimization methods

### 3.1    Convergence results of SGD

We consider the standard SGD method for solving the smooth optimization problem $\min_{\boldsymbol{x} \in \mathbb{R}^n} f(\boldsymbol{x})$, where the iteration of SGD is given by

$$\boldsymbol{x}^{k+1} = \boldsymbol{x}^k - \alpha_k \boldsymbol{g}^k. \tag{5}$$

Here, $\boldsymbol{g}^k$ denotes a stochastic approximation of the gradient $\nabla f(\boldsymbol{x}^k)$. We assume that each stochastic gradient $\boldsymbol{g}^k$ is $\mathcal{F}_{k+1}$-measurable and that the generated stochastic process $\{\boldsymbol{x}^k\}_{k \geq 0}$ is adapted to the filtration $\{\mathcal{F}_k\}_{k \geq 0}$. We consider the following standard assumptions:

(A.1)  The mapping $\nabla f : \mathbb{R}^n \to \mathbb{R}^n$ is Lipschitz continuous on $\mathbb{R}^n$ with modulus $\mathsf{L} > 0$.

(A.2)  The objective function $f$ is bounded from below on $\mathbb{R}^n$, i.e., there is $\bar{f}$ such that $f(\boldsymbol{x}) \geq \bar{f}$ for all $\boldsymbol{x} \in \mathbb{R}^n$.

(A.3)  Each oracle $\boldsymbol{g}^k$ defines an unbiased estimator of $\nabla f(\boldsymbol{x}^k)$, i.e., it holds that $\mathbb{E}[\boldsymbol{g}^k \mid \mathcal{F}_k] = \nabla f(\boldsymbol{x}^k)$ almost surely, and there exist $\mathsf{C}, \mathsf{D} \geq 0$ such that

$$\mathbb{E}[\|\boldsymbol{g}^k - \nabla f(\boldsymbol{x}^k)\|^2 \mid \mathcal{F}_k] \leq \mathsf{C}[f(\boldsymbol{x}^k) - \bar{f}] + \mathsf{D} \quad \text{almost surely} \quad \forall \, k \in \mathbb{N}.$$

(A.4)  The step sizes $\{\alpha_k\}_{k \geq 0}$ satisfy $\sum_{k=0}^{\infty} \alpha_k = \infty$ and $\sum_{k=0}^{\infty} \alpha_k^2 < \infty$.

We now derive the convergence of SGD below by setting $\boldsymbol{\Phi} \equiv \nabla f$ and $\mu_k \equiv \alpha_k$.

**Phase I: Verifying (P.1)–(P.2).** (A.1) verifies condition (P.1) with $\mathsf{L}_\Phi \equiv \mathsf{L}$. We now check (P.2). Using (A.2), (A.3), and a standard analysis for SGD gives the following recursion (see Appendix C.1 for the full derivation):

$$\mathbb{E}[f(\boldsymbol{x}^{k+1}) - \bar{f} \mid \mathcal{F}_k] \leq \left(1 + \frac{\mathsf{LC}\alpha_k^2}{2}\right)[f(\boldsymbol{x}^k) - \bar{f}] - \alpha_k \left(1 - \frac{\mathsf{L}\alpha_k}{2}\right) \|\nabla f(\boldsymbol{x}^k)\|^2 + \frac{\mathsf{LD}\alpha_k^2}{2}. \tag{6}$$

Taking total expectation, using (A.4), and applying the supermartingale convergence theorem (Theorem B.1) gives $\sum_{k=0}^{\infty} \alpha_k \mathbb{E}[\|\nabla f(\boldsymbol{x}^k)\|^2] < \infty$. Furthermore, the sequence $\{\mathbb{E}[f(\boldsymbol{x}^k)]\}_{k \geq 0}$ converges to some finite value. This verifies (P.2) with $a = 2$.

**Phase II: Verifying (P.3)–(P.4) for showing expected convergence.** For (P.3), we have by (5) and (A.3) that

$$\mathbb{E}[\|\boldsymbol{x}^{k+1} - \boldsymbol{x}^k\|^2] \leq \alpha_k^2 \mathbb{E}[\|\nabla f(\boldsymbol{x}^k)\|^2] + \mathsf{C}\alpha_k^2 \mathbb{E}[f(\boldsymbol{x}^k) - \bar{f}] + \mathsf{D}\alpha_k^2.$$

Due to the convergence of $\{\mathbb{E}[f(\boldsymbol{x}^k)]\}_{k \geq 0}$, there exists $\mathsf{F}$ such that $\mathbb{E}[f(\boldsymbol{x}^k) - \bar{f}] \leq \mathsf{F}$ for all $k$. Thus, condition (P.3) holds with $q = 2$, $\mathsf{A} = \mathsf{CF} + \mathsf{D}$, $p_1 = 2$, $\mathsf{B} = 1$, $p_2 = 2$, and $b = 2$. Condition (P.4) is verified by (A.4) and the previous parameters choices. Therefore, we can apply Theorem 2.1 to deduce $\mathbb{E}[\|\nabla f(\boldsymbol{x}^k)\|] \to 0$.

**Phase III: Verifying (P.3′)–(P.4′) for showing almost sure convergence.** For (P.3′), it follows from the update (5) that

$$\boldsymbol{x}^{k+1} = \boldsymbol{x}^k - \alpha_k(\boldsymbol{g}^k - \nabla f(\boldsymbol{x}^k)) - \alpha_k \nabla f(\boldsymbol{x}^k).$$

We have $p_1 = 1$, $\boldsymbol{A}_k = \boldsymbol{g}^k - \nabla f(\boldsymbol{x}^k)$, $p_2 = 1$, and $\boldsymbol{B}_k = \nabla f(\boldsymbol{x}^k)$. Using (A.2), (A.3), $\mathbb{E}[f(\boldsymbol{x}^k) - \bar{f}] \leq \mathsf{F}$, and choosing any $q = b > 0$ establishes (P.3′). As before, condition (P.4′) follows from (A.4) and the previous parameters choices. Applying Theorem 2.1 yields $\|\nabla f(\boldsymbol{x}^k)\| \to 0$ almost surely.

Finally, we summarize the above results in the following corollary.

**Corollary 3.1.** *Let us consider* SGD *(5) for smooth nonconvex optimization problems under (A.1)–(A.4). Then, we have* $\lim_{k \to \infty} \mathbb{E}[\|\nabla f(\boldsymbol{x}^k)\|] = 0$ *and* $\lim_{k \to \infty} \|\nabla f(\boldsymbol{x}^k)\| = 0$ *almost surely.*

## 3.2 Convergence results of random reshuffling

We now consider random reshuffling (RR) applied to problems with a finite sum structure

$$\min_{\boldsymbol{x} \in \mathbb{R}^n} f(\boldsymbol{x}) := \frac{1}{N} \sum_{i=1}^{N} f(\boldsymbol{x}, i),$$

where each component function $f(\cdot, i) : \mathbb{R}^n \to \mathbb{R}$ is supposed to be smooth. At iteration $k$, RR first generates a random permutation $\sigma^{k+1}$ of the index set $\{1, \ldots, N\}$. It then updates $\boldsymbol{x}^k$ to $\boldsymbol{x}^{k+1}$ through $N$ consecutive gradient descent-type steps by accessing and using the component gradients $\{\nabla f(\cdot, \sigma_1^{k+1}), \ldots, \nabla f(\cdot, \sigma_N^{k+1})\}$ sequentially. Specifically, one update-loop (epoch) of RR is given by

$$\tilde{\boldsymbol{x}}_0^k = \boldsymbol{x}^k, \quad \tilde{\boldsymbol{x}}_i^k = \tilde{\boldsymbol{x}}_{i-1}^k - \alpha_k \nabla f(\tilde{\boldsymbol{x}}_{i-1}^k, \sigma_i^{k+1}), \quad i = 1, \ldots, N, \quad \boldsymbol{x}^{k+1} = \tilde{\boldsymbol{x}}_N^k. \tag{7}$$

After one such loop, the step size $\alpha_k$ and the permutation $\sigma^{k+1}$ is updated accordingly; cf. [20, 30, 32]. We make the following standard assumptions:

(B.1) For all $i \in \{1, \ldots, N\}$, $f(\cdot, i)$ is bounded from below by some $\bar{f}$ and the gradient $\nabla f(\cdot, i)$ is Lipschitz continuous on $\mathbb{R}^n$ with modulus $\mathsf{L} > 0$.

(B.2) The step sizes $\{\alpha_k\}_{k \geq 0}$ satisfy $\sum_{k=0}^{\infty} \alpha_k = \infty$ and $\sum_{k=0}^{\infty} \alpha_k^3 < \infty$.

A detailed derivation of the steps shown in Subsection 2.1 for RR is deferred to Appendix D.2. Based on the discussion in Appendix D.2 and on Theorem 2.1, we obtain the following results for RR.

**Corollary 3.2.** *We consider* RR (7) *for smooth nonconvex optimization problems under* (B.1)–(B.2). *Then it holds that* $\lim_{k \to \infty} \mathbb{E}[\|\nabla f(\boldsymbol{x}^k)\|] = 0$ *and* $\lim_{k \to \infty} \|\nabla f(\boldsymbol{x}^k)\| = 0$ *almost surely.*

## 3.3 Convergence of the proximal stochastic gradient method

We consider the composite-type optimization problem

$$\min_{\boldsymbol{x} \in \mathbb{R}^n} \psi(\boldsymbol{x}) := f(\boldsymbol{x}) + \varphi(\boldsymbol{x}) \tag{8}$$

where $f : \mathbb{R}^n \to \mathbb{R}$ is a continuously differentiable function and $\varphi : \mathbb{R}^n \to (-\infty, \infty]$ is $\tau$-weakly convex (see Appendix E.1), proper, and lower semicontinuous. In this section, we want to apply our unified framework to study the convergence behavior of the well-known proximal stochastic gradient method (prox-SGD):

$$\boldsymbol{x}^{k+1} = \mathrm{prox}_{\alpha_k \varphi}(\boldsymbol{x}^k - \alpha_k \boldsymbol{g}^k), \tag{9}$$

where $\boldsymbol{g}^k \approx \nabla f(\boldsymbol{x}^k)$ is a stochastic approximation of $\nabla f(\boldsymbol{x}^k)$, $\{\alpha_k\}_{k \geq 0} \subseteq \mathbb{R}_+$ is a suitable step size sequence, and $\mathrm{prox}_{\alpha_k \varphi} : \mathbb{R}^n \to \mathbb{R}^n$, $\mathrm{prox}_{\alpha_k \varphi}(\boldsymbol{x}) := \mathrm{argmin}_{\boldsymbol{y} \in \mathbb{R}^n} \varphi(\boldsymbol{y}) + \frac{1}{2\alpha_k} \|\boldsymbol{x} - \boldsymbol{y}\|^2$ is the well-known proximity operator of $\varphi$.

### 3.3.1 Assumptions and preparations

We first recall several useful concepts from nonsmooth and variational analysis. For a function $h : \mathbb{R}^n \to (-\infty, \infty]$, the Fréchet (or regular) subdifferential of $h$ at the point $\boldsymbol{x}$ is given by

$$\partial h(\boldsymbol{x}) := \{\boldsymbol{g} \in \mathbb{R}^n : h(\boldsymbol{y}) \geq h(\boldsymbol{x}) + \langle \boldsymbol{g}, \boldsymbol{y} - \boldsymbol{x} \rangle + o(\|\boldsymbol{y} - \boldsymbol{x}\|) \text{ as } \boldsymbol{y} \to \boldsymbol{x}\},$$

see, e.g., [39, Chapter 8]. If $h$ is convex, then the Fréchet subdifferential coincides with the standard (convex) subdifferential. It is well-known that the associated first-order optimality condition for the composite problem (8) — $0 \in \partial \psi(\boldsymbol{x}) = \nabla f(\boldsymbol{x}) + \partial \varphi(\boldsymbol{x})$ — can be represented as a nonsmooth equation, [39, 21],

$$F_{\mathrm{nat}}^\alpha(\boldsymbol{x}) := \boldsymbol{x} - \mathrm{prox}_{\alpha \varphi}(\boldsymbol{x} - \alpha \nabla f(\boldsymbol{x})) = 0, \quad \alpha \in (0, \tau^{-1}),$$

where $F_{\mathrm{nat}}^\alpha$ denotes the so-called *natural residual*. The natural residual $F_{\mathrm{nat}}^\alpha$ is a common stationarity measure for the nonsmooth problem (8) and widely used in the analysis of proximal methods.

We will make the following assumptions on $f$, $\varphi$, and the stochastic oracles $\{\boldsymbol{g}^k\}_{k \geq 0}$:

(C.1) The function $f$ is bounded from below on $\mathbb{R}^n$, i.e., there is $\bar{f}$ such that $f(\boldsymbol{x}) \geq \bar{f}$ for all $\boldsymbol{x} \in \mathbb{R}^n$, and the gradient mapping $\nabla f$ is Lipschitz continuous (on $\mathbb{R}^n$) with modulus $\mathsf{L} > 0$.

(C.2) The function $\varphi$ is $\tau$-weakly convex, proper, lower semicontinuous, and bounded from below on $\mathrm{dom}\,\varphi$, i.e., we have $\varphi(\boldsymbol{x}) \geq \bar{\varphi}$ for all $\boldsymbol{x} \in \mathrm{dom}\,\varphi$.

(C.3) There exists $\mathsf{L}_\varphi > 0$ such that $\varphi(\boldsymbol{x}) - \varphi(\boldsymbol{y}) \leq \mathsf{L}_\varphi \|\boldsymbol{x} - \boldsymbol{y}\|$ for all $\boldsymbol{x}, \boldsymbol{y} \in \operatorname{dom} \varphi$.

(C.4) Each $\boldsymbol{g}^k$ defines an unbiased estimator of $\nabla f(\boldsymbol{x}^k)$, i.e., we have $\mathbb{E}[\boldsymbol{g}^k \mid \mathcal{F}_k] = \nabla f(\boldsymbol{x}^k)$ almost surely, and there exist $\mathsf{C}, \mathsf{D} \geq 0$ such that

$$\mathbb{E}[\|\boldsymbol{g}^k - \nabla f(\boldsymbol{x}^k)\|^2 \mid \mathcal{F}_k] \leq \mathsf{C}[f(\boldsymbol{x}^k) - \bar{f}] + \mathsf{D} \quad \text{almost surely} \quad \forall \, k \in \mathbb{N}.$$

(C.5) The step sizes $\{\alpha_k\}_{k\geq 0}$ satisfy $\sum_{k=0}^\infty \alpha_k = \infty$ and $\sum_{k=0}^\infty \alpha_k^2 < \infty$.

Here, we again assume that the generated stochastic processes $\{\boldsymbol{x}^k\}_{k\geq 0}$ is adapted to the filtration $\{\mathcal{F}_k\}_{k\geq 0}$. The assumptions (C.1), (C.2), (C.4), and (C.5) are fairly standard and broadly applicable. In particular, (C.1), (C.4), and (C.5) coincide with the conditions (A.1)–(A.4) used in the analysis of SGD. We continue with several remarks concerning condition (C.3).

**Remark 3.3.** Assumption (C.3) requires the mapping $\varphi$ to be Lipschitz continuous on its effective domain $\operatorname{dom} \varphi$. This condition holds in many important applications, e.g., when $\varphi$ is chosen as a norm or indicator function. Nonconvex examples satisfying (C.2) and (C.3) include, e.g., the minimax concave penalty (MCP) function [45], the smoothly clipped absolute deviation (SCAD) [15], or the student-t loss function. We refer to [4] and Appendix E.2 for further discussion.

### 3.3.2 Convergence results of prox-SGD

We now analyze the convergence of the random process $\{\boldsymbol{x}^k\}_{k\geq 0}$ generated by the stochastic algorithmic scheme (9). As pioneered in [11], we will use the Moreau envelope $\operatorname{env}_{\theta\psi}$,

$$\operatorname{env}_{\theta\psi} : \mathbb{R}^n \to \mathbb{R}, \quad \operatorname{env}_{\theta\psi}(\boldsymbol{x}) := \min_{\boldsymbol{y}\in\mathbb{R}^n} \psi(\boldsymbol{y}) + \frac{1}{2\theta}\|\boldsymbol{x} - \boldsymbol{y}\|^2, \tag{10}$$

as a smooth Lyapunov function to study the descent properties and convergence of prox-SGD.

We first note that the conditions (C.1) and (C.2) imply $\theta^{-1}$-weak convexity of $\psi$ for every $\theta \in (0, (\mathsf{L} + \tau)^{-1}]$. In this case, the Moreau envelope $\operatorname{env}_{\theta\psi}$ is a well-defined and continuously differentiable function with gradient $\nabla \operatorname{env}_{\theta\psi}(\boldsymbol{x}) = \frac{1}{\theta}(\boldsymbol{x} - \operatorname{prox}_{\theta\psi}(\boldsymbol{x}))$; see, e.g., [38, Theorem 31.5].

As shown in [13, 11], the norm of the Moreau envelope — $\|\nabla \operatorname{env}_{\theta\psi}(x)\|$ — defines an alternative stationarity measure for problem (8) that is equivalent to the natural residual if $\theta$ is chosen sufficiently small. A more explicit derivation of this connection is provided in Lemma E.1.

Next, we establish convergence of prox-SGD by setting $\boldsymbol{\Phi} \equiv \nabla \operatorname{env}_{\theta\psi}$ and $\mu_k \equiv \alpha_k$. Our analysis is based on the following two estimates which are verified in Appendix E.4 and Appendix E.5.

**Lemma 3.4.** Let $\{\boldsymbol{x}^k\}_{k\geq 0}$ be generated by prox-SGD and let the assumptions (C.1)–(C.4) be satisfied. Then, for $\theta \in (0, [3\mathsf{L} + \tau]^{-1})$ and all $k$ with $\alpha_k \leq \min\{\frac{1}{2\tau}, \frac{1}{2(\theta^{-1}-[\mathsf{L}+\tau])}\}$, it holds that

$$\mathbb{E}[\operatorname{env}_{\theta\psi}(\boldsymbol{x}^{k+1}) - \bar{\psi} \mid \mathcal{F}_k] \leq (1 + 4\mathsf{C}\theta^{-1}\alpha_k^2) \cdot [\operatorname{env}_{\theta\psi}(\boldsymbol{x}^k) - \bar{\psi}]$$
$$- \mathsf{L}\theta\alpha_k\|\nabla \operatorname{env}_{\theta\psi}(\boldsymbol{x}^k)\|^2 + 2\alpha_k^2(\mathsf{C}\mathsf{L}_\varphi^2 + \mathsf{D}\theta^{-1}), \tag{11}$$

almost surely, where $\bar{\psi} := \bar{f} + \bar{\varphi}$.

**Lemma 3.5.** Let $\{\boldsymbol{x}^k\}_{k\geq 0}$ be generated by prox-SGD and suppose that the assumptions (C.1)–(C.4) hold. Then, for $\theta \in (0, [\frac{4}{3}\mathsf{L} + \tau]^{-1})$ and all $k$ with $\alpha_k \leq \frac{1}{2\tau}$, we have almost surely

$$\mathbb{E}[\|\boldsymbol{x}^{k+1} - \boldsymbol{x}^k\|^2 \mid \mathcal{F}_k] \leq 8(2\mathsf{L} + \mathsf{C})\alpha_k^2 \cdot [\operatorname{env}_{\theta\psi}(\boldsymbol{x}^k) - \bar{\psi}] + 4(((2\mathsf{L} + \mathsf{C})\theta + 1)\mathsf{L}_\varphi^2 + \mathsf{D})\alpha_k^2. \tag{12}$$

**Phase I: Verifying (P.1)–(P.2).** In [21, Corollary 3.4], it is shown that the gradient of the Moreau envelope is Lipschitz continuous with modulus $\mathsf{L}_e := \max\{\theta^{-1}, (1 - [\mathsf{L} + \tau]\theta)^{-1}[\mathsf{L} + \tau]\}$ for all $\theta \in (0, [\mathsf{L} + \tau]^{-1})$. Thus, condition (P.1) is satisfied.

Furthermore, due to $\alpha_k \to 0$ and choosing $\theta \in (0, [3\mathsf{L} + \tau]^{-1})$, the estimate (11) in Lemma 3.4 holds for all $k$ sufficiently large. Consequently, due to $\operatorname{env}_{\theta\psi}(\boldsymbol{x}) \geq \psi(\operatorname{prox}_{\theta\psi}(\boldsymbol{x})) \geq \bar{\psi}$ and (C.5), Theorem B.1 is applicable and upon taking total expectation, $\{\mathbb{E}[\operatorname{env}_{\theta\psi}(\boldsymbol{x}^k)]\}_{k\geq 0}$ converges to some $\mathsf{E} \in \mathbb{R}$. In addition, the sequence $\{\operatorname{env}_{\theta\psi}(\boldsymbol{x}^k)\}_{k\geq 0}$ converges almost surely to some random variable $e^\star$ and we have $\sum_{k=0}^\infty \alpha_k \mathbb{E}[\|\nabla \operatorname{env}_{\theta\psi}(\boldsymbol{x}^k)\|^2] < \infty$. This verifies condition (P.2) with $a = 2$.

**Phase II: Verifying (P.3)–(P.4) for showing convergence in expectation.** Assumptions (C.1)–(C.5) and Lemma 3.5 allow us to establish the required bound stated in (P.3). Specifically, taking total

expectation in (12), we have

$$\mathbb{E}[\|\boldsymbol{x}^{k+1} - \boldsymbol{x}^k\|^2] \leq 8(2\mathsf{L} + \mathsf{C})\alpha_k^2 \cdot \mathbb{E}[\text{env}_{\theta\psi}(\boldsymbol{x}^k) - \bar{\psi}] + 4(((2\mathsf{L} + \mathsf{C})\theta + 1)\mathsf{L}_\varphi^2 + \mathsf{D})\alpha_k^2$$

for all $k$ sufficiently large. Due to $\mathbb{E}[\text{env}_{\theta\psi}(\boldsymbol{x}^k)] \to \mathsf{E}$, there exists $\mathsf{F}$ such that $\mathbb{E}[\text{env}_{\theta\psi}(\boldsymbol{x}^k) - \bar{\psi}] \leq \mathsf{F}$ for all $k$. Hence, (P.3) holds with $q = 2$, $\mathsf{A} = 8(2\mathsf{L} + \mathsf{C})\mathsf{F} + 4(((2\mathsf{L} + \mathsf{C})\theta + 1)\mathsf{L}_\varphi^2 + \mathsf{D})$, $p_1 = 2$, and $\mathsf{B} = 0$. The property (P.4) easily follows from (C.5) and the parameter choices. Consequently, using Theorem 2.1, we can infer $\mathbb{E}[\|\nabla\text{env}_{\theta\psi}(\boldsymbol{x}^k)\|] \to 0$.

**Phase III: Verifying (P.3′)–(P.4′) for showing almost sure convergence.** We follow the construction in (4) and set $\boldsymbol{A}_k = \alpha_k^{-1}(\boldsymbol{x}^{k+1} - \boldsymbol{x}^k - \mathbb{E}[\boldsymbol{x}^{k+1} - \boldsymbol{x}^k \mid \mathcal{F}_k])$, $\boldsymbol{B}_k = \alpha_k^{-1}\mathbb{E}[\boldsymbol{x}^{k+1} - \boldsymbol{x}^k \mid \mathcal{F}_k]$, and $p_1, p_2 = 1$. Clearly, we have $\mathbb{E}[\boldsymbol{A}_k \mid \mathcal{F}_k] = 0$ and based on the previous results in **Phase II**, we can show $\mathbb{E}[\|\boldsymbol{x}^{k+1} - \boldsymbol{x}^k\|^2] = \mathcal{O}(\alpha_k^2)$ which establishes boundedness of $\{\mathbb{E}[\|\boldsymbol{A}_k\|^2]\}_{k\geq 0}$. Similarly, for $\boldsymbol{B}_k$ and by Lemma 3.5 and Jensen's inequality, we obtain

$$\|\boldsymbol{B}_k\|^2 \leq \alpha_k^{-2}\mathbb{E}[\|\boldsymbol{x}^{k+1} - \boldsymbol{x}^k\|^2 \mid \mathcal{F}_k] \leq 8(2\mathsf{L} + \mathsf{C}) \cdot [\text{env}_{\theta\psi}(\boldsymbol{x}^k) - \bar{\psi}] + \mathcal{O}(1).$$

Due to $\text{env}_{\theta\psi}(\boldsymbol{x}^k) \to e^\star$ almost surely, this shows $\limsup_{k\to\infty} \|\boldsymbol{B}_k\|^2 < \infty$ almost surely. Hence, all requirements in (P.3′) are satisfied with $q = 2$ and $b = 0$. Moreover, it is easy to see that property (P.4′) also holds in this case. Overall, Theorem 2.1 implies $\|\nabla\text{env}_{\theta\psi}(\boldsymbol{x}^k)\| \to 0$ almost surely.

As mentioned, it is possible to express the obtained convergence results in terms of the natural residual $F_{\text{nat}} = F_{\text{nat}}^1$, see, e.g., Lemma E.1. We summarize our observations in the following corollary.

**Corollary 3.6.** *Let us consider* prox-SGD (9) *for the composite problem* (8) *under* (C.1)–(C.5)*. Then, we have* $\lim_{k\to\infty} \mathbb{E}[\|F_{\text{nat}}(\boldsymbol{x}^k)\|] = 0$ *and* $\lim_{k\to\infty} \|F_{\text{nat}}(\boldsymbol{x}^k)\| = 0$ *almost surely.*

**Remark 3.7.** As a byproduct, Lemma 3.4 also leads to an expected iteration complexity result of prox-SGD by using the ABC condition (C.4) rather than the standard bounded variance assumption. This is a nontrivial extension of [11, Corollary 3.6]. We provide a full derivation in Appendix E.6.

## 3.4 Convergence of stochastic model-based methods

In this section, we consider the convergence of stochastic model-based methods for nonsmooth weakly convex optimization problems

$$\min_{\boldsymbol{x}\in\mathbb{R}^n} \psi(\boldsymbol{x}) := f(\boldsymbol{x}) + \varphi(\boldsymbol{x}) = \mathbb{E}_{\xi\sim D}[f(\boldsymbol{x}, \xi)] + \varphi(\boldsymbol{x}), \tag{13}$$

where both $f$ and $\varphi$ are assumed to be (nonsmooth) weakly convex functions and $\psi$ is lower bounded, i.e., $\psi(\boldsymbol{x}) \geq \bar{\psi}$ for all $\boldsymbol{x} \in \text{dom } \varphi$. Classical stochastic optimization methods — including proximal stochastic subgradient, stochastic proximal point, and stochastic prox-linear methods — for solving (13) are unified by the stochastic model-based methods (SMM) [14, 11]:

$$\boldsymbol{x}^{k+1} = \text{argmin}_{\boldsymbol{x}\in\mathbb{R}^n} f_{\boldsymbol{x}^k}(\boldsymbol{x}, \xi^k) + \varphi(\boldsymbol{x}) + \frac{1}{2\alpha_k}\|\boldsymbol{x} - \boldsymbol{x}^k\|^2, \tag{14}$$

where $f_{\boldsymbol{x}^k}(\boldsymbol{x}, \xi^k)$ is a stochastic approximation of $f$ around $\boldsymbol{x}^k$ using the sample $\xi^k$; see Appendix F.1 for descriptions of three major types of SMM. Setting $\mathcal{F}_k := \sigma(\xi^0, \ldots, \xi^{k-1})$, it is easy to see that $\{\boldsymbol{x}^k\}_{k\geq 0}$ is adapted to $\{\mathcal{F}_k\}_{k\geq 0}$. We analyze convergence of SMM under the following assumptions.

(D.1) The stochastic approximation function $f_{\boldsymbol{x}}$ satisfies a one-sided accuracy property, i.e., we have $\mathbb{E}_\xi[f_{\boldsymbol{x}}(\boldsymbol{x}, \xi)] = f(\boldsymbol{x})$ for all $\boldsymbol{x} \in U$ and

$$\mathbb{E}_\xi[f_{\boldsymbol{x}}(\boldsymbol{y}, \xi) - f(\boldsymbol{y})] \leq \frac{\tau}{2}\|\boldsymbol{x} - \boldsymbol{y}\|^2 \quad \forall \boldsymbol{x}, \boldsymbol{y} \in U,$$

where $U$ is an open convex set containing $\text{dom } \varphi$.

(D.2) The function $\boldsymbol{y} \mapsto f_{\boldsymbol{x}}(\boldsymbol{y}, \xi) + \varphi(\boldsymbol{y})$ is $\eta$-weakly convex for all $\boldsymbol{x} \in U$ and almost every $\xi$.

(D.3) There exists $\mathsf{L} > 0$ such that the stochastic approximation function $f_{\boldsymbol{x}}$ satisfies

$$f_{\boldsymbol{x}}(\boldsymbol{x}, \xi) - f_{\boldsymbol{x}}(\boldsymbol{y}, \xi) \leq \mathsf{L}\|\boldsymbol{x} - \boldsymbol{y}\| \quad \forall \boldsymbol{x}, \boldsymbol{y} \in U, \quad \text{and almost every } \xi.$$

(D.4) The function $\varphi$ is $\mathsf{L}_\varphi$-Lipschitz continuous.

(D.5) The step sizes $\{\alpha_k\}_{k\geq 0}$ satisfy $\sum_{k=0}^\infty \alpha_k = \infty$ and $\sum_{k=0}^\infty \alpha_k^2 < \infty$.

Assumptions (D.1), (D.2), (D.3) are standard for analyzing SMM and identical to that of [11]. (D.5) is convention for stochastic methods. Assumption (D.4) mimics (C.3); see Remark 3.3 for discussions.

We now derive the convergence of SMM below by setting $\boldsymbol{\Phi} \equiv \nabla\mathrm{env}_{\theta\psi}$ and $\mu_k \equiv \alpha_k$. Our derivation is based on the following two estimates, in which the proof of Lemma 3.9 is given in Appendix F.2.

**Lemma 3.8** (Theorem 4.3 of [11]). *Let $\theta \in (0, (\tau + \eta)^{-1})$ and $\alpha_k < \theta$ be given. Then, we have*

$$\mathbb{E}[\mathrm{env}_{\theta\psi}(\boldsymbol{x}^{k+1}) \mid \mathcal{F}_k] \leq \mathrm{env}_{\theta\psi}(\boldsymbol{x}^k) - \frac{(1 - [\tau + \eta]\theta)\alpha_k}{2(1 - \eta\alpha_k)}\|\nabla\mathrm{env}_{\theta\psi}(\boldsymbol{x}^k)\|^2 + \frac{2\mathsf{L}^2\alpha_k^2}{(1 - \eta\alpha_k)(\theta - \alpha_k)}.$$

**Lemma 3.9.** *For all $k$ with $\alpha_k \leq 1/(2\eta)$, it holds that*

$$\mathbb{E}[\|\boldsymbol{x}^{k+1} - \boldsymbol{x}^k\|^2 \mid \mathcal{F}_k] \leq (16(\mathsf{L} + \mathsf{L}_\varphi)^2 + 8\mathsf{L}^2)\alpha_k^2.$$

**Phase I: Verifying (P.1)–(P.2).** As before, [21, Corollary 3.4] implies that the mapping $\nabla\mathrm{env}_{\theta\psi}$ is Lipschitz continuous for all $\theta \in (0, (\tau + \eta)^{-1})$ Hence, condition (P.1) is satisfied. Using $\alpha_k \to 0$, we can apply Theorem B.1 to the recursion obtained in Lemma 3.8 for all $k$ sufficiently large and it follows $\sum_{k=0}^{\infty} \alpha_k\mathbb{E}[\|\nabla\mathrm{env}_{\theta\psi}(\boldsymbol{x}^k)\|^2] < \infty$. Thus, condition (P.2) holds with $a = 2$.

**Phase II: Verifying (P.3)–(P.4) for showing convergence in expectation.** Taking total expectation in Lemma 3.9 verifies condition (P.3) with $q = 2$, $\mathsf{A} = (16(\mathsf{L} + \mathsf{L}_\varphi)^2 + 8\mathsf{L}^2)$, $p_1 = 2$, $\mathsf{B} = 0$. Moreover, condition (P.4) is true by assumption (D.5) and the previous parameters choices. Thus, applying Theorem 2.1 gives $\mathbb{E}[\|\nabla\mathrm{env}_{\theta\psi}(\boldsymbol{x}^k)\|] \to 0$.

**Phase III: Verifying (P.3′)–(P.4′) for showing almost sure convergence.** As in (4), we can set $\boldsymbol{A}_k = \alpha_k^{-1}(\boldsymbol{x}^{k+1} - \boldsymbol{x}^k - \mathbb{E}[\boldsymbol{x}^{k+1} - \boldsymbol{x}^k \mid \mathcal{F}_k])$, $\boldsymbol{B}_k = \alpha_k^{-1}\mathbb{E}[\boldsymbol{x}^{k+1} - \boldsymbol{x}^k \mid \mathcal{F}_k]$. Applying Lemma 3.9 and utilizing Jensen's inequality, we have $\mathbb{E}[\boldsymbol{A}_k \mid \mathcal{F}_k] = 0$, $\mathbb{E}[\|\boldsymbol{A}_k\|^2] \leq (4/\alpha_k^2)\mathbb{E}[\|\boldsymbol{x}^{k+1} - \boldsymbol{x}^k\|^2] \leq 4(16(\mathsf{L} + \mathsf{L}_\varphi)^2 + 8\mathsf{L}^2)$ and $\|\boldsymbol{B}_k\|^2 \leq 16(\mathsf{L} + \mathsf{L}_\varphi)^2 + 8\mathsf{L}^2$. Thus, condition (P.3′) is satisfied with $p_1 = p_2 = 1$, $q = 2$. Assumption (D.5), together with the previous parameter choices verifies condition (P.4′) and hence, applying Theorem 2.1 yields $\|\nabla\mathrm{env}_{\theta\psi}(\boldsymbol{x}^k)\| \to 0$ almost surely.

Summarizing this discussion, we obtain the following convergence results for SMM.

**Corollary 3.10.** *We consider the family of stochastic model-based methods* (14) *for the optimization problem* (13) *under assumptions (D.1)–(D.5). Let $\{\boldsymbol{x}^k\}_{k\geq 0}$ be a generated sequence. Then, we have $\lim_{k\to\infty} \mathbb{E}[\|\nabla\mathrm{env}_{\theta\psi}(\boldsymbol{x}^k)\|] = 0$ and $\lim_{k\to\infty} \|\nabla\mathrm{env}_{\theta\psi}(\boldsymbol{x}^k)\| = 0$ almost surely.*

**Remark 3.11.** The results presented in Corollary 3.10 also hold under certain extended settings. In fact, we can replace (D.3) by a slightly more general Lipschitz continuity assumption on $f$. Moreover, it is possible to establish convergence in the case where $f$ is not Lipschitz continuous but has Lipschitz continuous gradient, which is particularly useful when we apply stochastic proximal point method for smooth $f$. A more detailed derivation and discussion of such extensions is deferred to Appendix F.3.

### 3.5 Related work and discussion

**SGD and RR.** The literature for SGD is extremely rich and several connected and recent works have been discussed in Section 1. Our result in Corollary 3.1 unifies many of the existing convergence analyses of SGD and is based on the general ABC condition (A.3) (see [23, 24, 19] for comparison) rather than on the standard bounded variance assumption. Our expected convergence result generalizes the one in [6] using much weaker assumptions. Our results for RR are in line with the recent theoretical observations in [30, 32, 25]. In particular, Corollary 3.2 recovers the almost sure convergence result shown in [25], while the expected convergence result appears to be new.

**Prox-SGD and SMM.** The work [11] established one of the first complexity results for prox-SGD using the Moreau envelope. Under a bounded variance assumption ($\mathsf{C} = 0$ in condition (C.4)) and for general nonconvex and smooth $f$, the authors showed $\mathbb{E}[\|\nabla\mathrm{env}_{\theta\psi}(\boldsymbol{x}^{\bar{k}})\|^2] = \mathcal{O}((T + 1)^{-1/2})$, where $\boldsymbol{x}^{\bar{k}}$ is sampled uniformly from the past $T + 1$ iterates $\boldsymbol{x}^0, \ldots, \boldsymbol{x}^T$. As mentioned, this result cannot be easily extended to the asymptotic convergence results discussed in this paper. Earlier studies of prox-SGD for nonconvex $f$ and $\mathsf{C} = 0$ include [18] where convergence of prox-SGD is established if the variance parameter $\mathsf{D} = \mathsf{D}_k \to 0$ vanishes as $k \to \infty$. This can be achieved by progressively increasing the size of the selected mini-batches or via variance reduction techniques as in prox-SVRG and prox-SAGA, see [35]. The question whether prox-SGD can converge and whether the accumulation points of the iterates $\{\boldsymbol{x}^k\}_{k\geq 0}$ correspond to stationary points was only addressed recently in [27]. The authors use a differential inclusion approach to study convergence of prox-SGD. However, additional compact constraints $\boldsymbol{x} \in \mathcal{X}$ have to be introduced in the model (8) to guarantee sure boundedness of $\{\boldsymbol{x}^k\}_{k\geq 0}$ and applicability of the differential inclusion techniques. Lipschitz

continuity of $\varphi$ also appears as an essential requirement in [27, Theorem 5.4]. The analyses in [14, 12] establish asymptotic convergence guarantees for SMM. However, both works require a priori (almost) sure boundedness of $\{x^k\}_{k \geq 0}$ and a density / Sard-type condition in order to show convergence. We refer to [16] for an extension of the results in [27, 12] to prox-SGD in Hilbert spaces. By contrast, our convergence framework allows to complement these differential inclusion-based results and — for the first time — fully removes any stringent boundedness assumption on $\{x^k\}_{k \geq 0}$. Instead, our analysis relies on more transparent assumptions that are verifiable and common in stochastic optimization and machine learning. In summary, we are now able to claim: prox-SGD *and* SMM *converge under standard stochastic conditions if $\varphi$ is Lipschitz continuous*. In the easier convex case, analogous results have been obtained, e.g., in [18, 1, 40].

We provide an overview of several related and representative results in Table 1 in Appendix G.

## 4   Conclusion

In this work, we provided a novel convergence framework that allows to derive expected and almost sure convergence results for a vast class of stochastic optimization methods under state-of-the-art assumptions and in a unified way. We specified the steps on how to utilize our theorem in order to establish convergence results for a given stochastic algorithm. As concrete examples, we applied our theorem to derive asymptotic convergence guarantees for SGD, RR, prox-SGD, and SMM. To our surprise, some of the obtained results appear to be new and provide new insights into the convergence behavior of some well-known and standard stochastic methodologies. These applications revealed that our unified theorem can serve as a plugin-type tool with the potential to facilitate the convergence analysis of a wide class of stochastic optimization methods.

Finally, it is important to investigate in which situations our convergence results in terms of the stationarity measure $\Phi$ can be strengthened — say to almost sure convergence guarantees for the iterates $\{x^k\}_{k \geq 0}$. We plan to consider such a possible extension in future work.

## Acknowledgments and Disclosure of Funding

The authors would like to thank the Area Chair and anonymous reviewers for their detailed and constructive comments, which have helped greatly to improve the quality and presentation of the manuscript. In addition, we would like to thank Michael Ulbrich for valuable feedback and comments on an earlier version of this work.

X. Li was partially supported by the National Natural Science Foundation of China (NSFC) under Grant No. 12201534 and 72150002, by the Shenzhen Science and Technology Program under Grant No. RCBS20210609103708017, and by the Shenzhen Institute of Artificial Intelligence and Robotics for Society (AIRS) under Grant No. AC01202101108. A. Milzarek was partly supported by the National Natural Science Foundation of China (NSFC) – Foreign Young Scholar Research Fund Project (RFIS) under Grant No. 12150410304 and by the Fundamental Research Fund – Shenzhen Research Institute of Big Data (SRIBD) Startup Fund JCYJ-AM20190601.

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
