# Contents

# A Proof of Theorem 2.1

**Part I: Proof of item (i).** First, combining $\sum_{k=0}^{\infty} \mu_k = \infty$ in (P.4) and (P.2), we can infer $\liminf_{k\to\infty} \mathbb{E}[\|\boldsymbol{\Phi}(\boldsymbol{x}^k)\|^a] = 0$. Let us assume that $\{\mathbb{E}[\|\boldsymbol{\Phi}(\boldsymbol{x}^k)\|]\}_{k\geq 0}$ does not converge to zero. Then, there exist $\delta > 0$ and two subsequences $\{\ell_t\}_{t\geq 0}$ and $\{u_t\}_{t\geq 0}$ such that $\ell_t < u_t \leq \ell_{t+1}$ and

$$\mathbb{E}[\|\boldsymbol{\Phi}(\boldsymbol{x}^{\ell_t})\|] \geq 2\delta, \quad \mathbb{E}[\|\boldsymbol{\Phi}(\boldsymbol{x}^{u_t})\|^a] \leq \delta^a, \quad \text{and} \quad \mathbb{E}[\|\boldsymbol{\Phi}(\boldsymbol{x}^k)\|^a] > \delta^a \tag{15}$$

for all $\ell_t < k < u_t$ and $t \in \mathbb{N}$. Combining (P.2) and (15) and applying Jensen's inequality ($a \geq 1$), this yields

$$\infty > \sum_{t=0}^{\infty} \sum_{k=\ell_t}^{u_t-1} \mu_k \, \mathbb{E}[\|\boldsymbol{\Phi}(\boldsymbol{x}^k)\|^a] \geq \sum_{t=0}^{\infty} \left[ \sum_{k=\ell_t+1}^{u_t-1} \mu_k \delta^a + \mu_{\ell_t} (\mathbb{E}[\|\boldsymbol{\Phi}(\boldsymbol{x}^{\ell_t})\|])^a \right] \geq \delta^a \sum_{t=0}^{\infty} \sum_{k=\ell_t}^{u_t-1} \mu_k,$$

which immediately implies $\beta_t := \sum_{k=\ell_t}^{u_t-1} \mu_k \to 0$. Since $\{\mu_k\}_{k\geq 0}$ is bounded, there exists $\bar{\mu}$ such that $\mu_k \leq \bar{\mu}$ for all $k$. For any $p \geq 1$, this further implies $\mu_k^p \leq \bar{\mu}^{p-1}\mu_k$ for all $k$. Using Hölder's and Jensen's inequality, $q \geq 1$, and (P.3), we have

$$
\begin{aligned}
\mathbb{E}[\|\boldsymbol{x}^{u_t} - \boldsymbol{x}^{\ell_t}\|] &\leq \sum_{k=\ell_t}^{u_t-1} \mathbb{E}[\|\boldsymbol{x}^{k+1} - \boldsymbol{x}^k\|] \leq \left[\sum_{k=\ell_t}^{u_t-1} \mu_k\right]^{1-\frac{1}{q}} \left[\sum_{k=\ell_t}^{u_t-1} \mu_k^{-(q-1)} \mathbb{E}[\|\boldsymbol{x}^{k+1} - \boldsymbol{x}^k\|^q]\right]^{\frac{1}{q}} \\
&\leq \beta_t^{1-\frac{1}{q}} \left[\sum_{k=\ell_t}^{u_t-1} \mathsf{A}\mu_k^{p_1-(q-1)} + \mathsf{B}\mu_k^{p_2-(q-1)} \mathbb{E}[\|\boldsymbol{\Phi}(\boldsymbol{x}^k)\|^b]\right]^{\frac{1}{q}} \\
&\leq \beta_t^{1-\frac{1}{q}} \left[\sum_{k=\ell_t}^{u_t-1} \mathsf{A}\bar{\mu}^{p_1-q}\mu_k + \mathsf{B}\bar{\mu}^{p_2-q}\mu_k \mathbb{E}[\|\boldsymbol{\Phi}(\boldsymbol{x}^k)\|^b]\right]^{\frac{1}{q}} \\
&= \beta_t^{1-\frac{1}{q}} \left[\mathsf{A}\bar{\mu}^{p_1-q}\beta_t + \mathsf{B}\bar{\mu}^{p_2-q}\sum_{k=\ell_t}^{u_t-1} \mu_k \mathbb{E}[\|\boldsymbol{\Phi}(\boldsymbol{x}^k)\|^b]\right]^{\frac{1}{q}},
\end{aligned}
\tag{16}
$$

where we have utilized $p_1, p_2 \geq q$ in the third line. We first consider the case $a > b$. We can apply Jensen's and Hölder's inequality to obtain

$$
\begin{aligned}
\sum_{k=\ell_t}^{u_t-1} \mu_k \mathbb{E}[\|\boldsymbol{\Phi}(\boldsymbol{x}^k)\|^b] &\leq \sum_{k=\ell_t}^{u_t-1} \mu_k \mathbb{E}[\|\boldsymbol{\Phi}(\boldsymbol{x}^k)\|^a]^{\frac{b}{a}} \\
&\leq \left[\sum_{k=\ell_t}^{u_t-1} \mu_k\right]^{\frac{a-b}{a}} \left[\sum_{k=\ell_t}^{u_t-1} \mu_k \, \mathbb{E}[\|\boldsymbol{\Phi}(\boldsymbol{x}^k)\|^a]\right]^{\frac{b}{a}} \\
&= \beta_t^{\frac{a-b}{a}} \left[\sum_{k=\ell_t}^{u_t-1} \mu_k \, \mathbb{E}[\|\boldsymbol{\Phi}(\boldsymbol{x}^k)\|^a]\right]^{\frac{b}{a}}.
\end{aligned}
\tag{17}
$$

Plugging (17) into (16) and invoking (P.2) and $\beta_t \to 0$, we have $\mathbb{E}[\|\boldsymbol{x}^{u_t} - \boldsymbol{x}^{\ell_t}\|] \to 0$ as $t \to \infty$. We now consider the case $a = b$. Let us introduce $E_k := \sum_{i=0}^{k-1} \mu_i \, \mathbb{E}[\|\boldsymbol{\Phi}(\boldsymbol{x}^i)\|^a]$. By (P.2), the sequence $\{E_k\}_{k\geq 1}$ converges and hence, we have $E_{u_t} - E_{\ell_t} \to 0$ as $t \to \infty$. It then follows from (16) that

$$\mathbb{E}[\|\boldsymbol{x}^{u_t} - \boldsymbol{x}^{\ell_t}\|] \leq \beta_t^{1-\frac{1}{q}} [\mathsf{A}\beta_t + \mathsf{B}[E_{u_t} - E_{\ell_t}]]^{\frac{1}{q}} \to 0 \quad \text{as} \quad t \to \infty.$$

Together, this establishes $\mathbb{E}[\|\boldsymbol{x}^{u_t} - \boldsymbol{x}^{\ell_t}\|] \to 0$ as $t \to \infty$ in both cases. However, by the Lipschitz continuity of $\boldsymbol{\Phi}$ in (P.1), $a \geq 1$ in (P.4), and the construction (15), we have

$$\delta \leq \mathbb{E}[\|\boldsymbol{\Phi}(\boldsymbol{x}^{\ell_t})\|] - \mathbb{E}[\|\boldsymbol{\Phi}(\boldsymbol{x}^{u_t})\|] \leq \mathbb{E}[\|\boldsymbol{\Phi}(\boldsymbol{x}^{u_t}) - \boldsymbol{\Phi}(\boldsymbol{x}^{\ell_t})\|] \leq \mathsf{L}_{\boldsymbol{\Phi}}\mathbb{E}[\|\boldsymbol{x}^{u_t} - \boldsymbol{x}^{\ell}\|]. \tag{18}$$

By letting $t \to \infty$ in (18), we get a contradiction. This concludes the proof of item (i).

**Part II: Proof of item (ii).** In order to control the stochastic behavior of the error terms $\boldsymbol{A}_k$ and to establish the almost sure convergence of the sequence $\{\|\boldsymbol{\Phi}(\boldsymbol{x}^k)\|\}_{k\geq 0}$, we will utilize several results from martingale theory.

**Definition A.1** (Martingale). *Let $(\Omega, \mathcal{U}, \mathbb{P})$ be a probability space and $\{\mathcal{U}_k\}_{k\geq 0}$ a family of increasing sub-$\sigma$-fields of $\mathcal{U}$. A random process $\{\boldsymbol{M}_k\}_{k\geq 0}$ defined on this probability space is said to be a*

*martingale with respect to the family $\{\mathcal{U}_k\}_{k\geq 0}$ (or an $\{\mathcal{U}_k\}_{k\geq 0}$-martingale) if each $M_k$ is integrable and $\mathcal{U}_k$-measurable and we have $\mathbb{E}[M_{k+1} \mid \mathcal{U}_k] = M_k$ a.s. for all $k$.*

Next, we state a standard convergence theorem for vector-valued martingales, see, e.g., [43, Theorem 5.2.22 and Section 5.3] or [7, Theorem 5.14].

**Theorem A.2** (Martingale Convergence Theorem). *Let $\{M_k\}_{k\geq 0}$ be a given vector-valued $\{\mathcal{U}_k\}_{k\geq 0}$-martingale as specified in Definition A.1. If $\sup_k \mathbb{E}[\|M_k\|] < \infty$, then $\{M_k\}_{k\geq 0}$ converges almost surely to an integrable random vector $M$.*

**Step (a): Analysis of the error terms $\{A_k\}_{k\geq 0}$.** Let us define $M_k := \sum_{i=0}^{k-1} \mu_i^{p_1} A_i$. Then, it holds that

$$\mathbb{E}[M_{k+1} \mid \mathcal{F}_k] = \sum_{i=0}^{k-1} \mu_i^{p_1} A_i + \mu_k^{p_1} \mathbb{E}[A_k \mid \mathcal{F}_k] = M_k,$$

i.e., $\{M_k\}_{k\geq 0}$ defines a martingale adapted to the filtration $\{\mathcal{F}_k\}_{k\geq 0}$. In addition, inductively, we obtain

$$\mathbb{E}[\|M_k\|^2] = \mathbb{E}[\|M_{k-1}\|^2] + 2\mu_{k-1}^{p_1}\mathbb{E}[\langle M_{k-1}, A_{k-1}\rangle] + \mu_{k-1}^{2p_1}\mathbb{E}[\|A_{k-1}\|^2]$$

$$= \cdots = \sum_{i=0}^{k-1} \mu_i^{2p_1}\mathbb{E}[\|A_i\|^2] \leq \sum_{i=0}^{k-1} \mu_i^{2p_1}\mathbb{E}[\|A_i\|^q]^{\frac{2}{q}} \leq \mathsf{A}^{\frac{2}{q}}\sum_{i=0}^{k-1}\mu_i^{2p_1},$$

where we utilized Jensen's inequality, the concavity of the mapping $x \mapsto x^{\frac{2}{q}}$, and $\mathbb{E}[\|A_i\|^q] \leq \mathsf{A}$ in (P.3′). Hence, by (P.4′) and Jensen's inequality, we can infer $\sup_k \mathbb{E}[\|M_k\|] \leq \mathsf{A}^{\frac{1}{q}}[\sum_{i=0}^{\infty}\mu_i^{2p_1}]^{\frac{1}{2}} < \infty$. Theorem A.2 then implies that $\{M_k\}_{k\geq 0}$ converges almost surely to some integrable random vector $M$.

Next, we establish almost sure convergence of $\{\|\Phi(x^k)\|\}_{k\geq 0}$. Our derivation generally mimics the proof of item (i), but uses sample-based arguments.

**Step (b): Almost sure convergence.** First, applying the monotone convergence theorem to (P.2) gives $\mathbb{E}[\sum_{k=0}^{\infty}\mu_k\|\Phi(x^k)\|^a] = \lim_{T\to\infty}\sum_{k=0}^{T}\mu_k\mathbb{E}[\|\Phi(x^k)\|^a] < \infty$, which immediately implies $\sum_{k=0}^{\infty}\mu_k\|\Phi(x^k)\|^a < \infty$ almost surely. This, together with $\sum_{k=0}^{\infty}\mu_k = \infty$, yields $\liminf_{k\to\infty}\|\Phi(x^k)\| = 0$ almost surely. We now consider an arbitrary sample $\omega \in \mathcal{M}$ where

$$\mathcal{M} := \left\{\omega : \sum_{k=0}^{\infty}\mu_k\|\Phi(x^k(\omega))\|^a < \infty, \quad \lim_{k\to\infty}M_k(\omega) = M(\omega),\right.$$

$$\left.\text{and} \quad \limsup_{k\to\infty}\frac{\|B_k(\omega)\|^q}{1 + \|\Phi(x^k(\omega))\|^b} < \infty\right\}.$$

Our preceding discussion implies that the event $\mathcal{M}$ occurs with probability 1 and it holds that $\liminf_{k\to\infty}\|\Phi(x^k(\omega))\| = 0$. Let us assume that $\{\|\Phi(x^k(\omega))\|\}_{k\geq 0}$ does not converge to zero. Then, there exist $\delta > 0$ and two subsequences $\{\ell_t\}_{t\geq 0}$ and $\{u_t\}_{t\geq 0}$ such that $\ell_t < u_t \leq \ell_{t+1}$ and

$$\|\Phi(x^{\ell_t}(\omega))\| \geq 2\delta, \quad \|\Phi(x^{u_t}(\omega))\| \leq \delta, \quad \text{and} \quad \|\Phi(x^k(\omega))\| > \delta \tag{19}$$

for all $\ell_t < k < u_t$ and $t \in \mathbb{N}$. (Notice that in contrast to the proof of item (i), the sequences $\{\ell_t\}_{t\geq 0}$, $\{u_t\}_{t\geq 0}$, and $\delta$ will now generally depend on the selected sample $\omega$). Due to $\omega \in \mathcal{M}$, this yields

$$\infty > \sum_{t=0}^{\infty}\sum_{k=\ell_t}^{u_t-1}\mu_k\|\Phi(x^k(\omega))\|^a \geq \delta^a\sum_{t=0}^{\infty}\sum_{k=\ell_t}^{u_t-1}\mu_k,$$

which again implies $\beta_t := \sum_{k=\ell_t}^{u_t-1}\mu_k \to 0$. Furthermore, there exist $K \in \mathbb{N}$ and $B \in \mathbb{R}$ such that $\|B_k(\omega)\| \leq B(1 + \|\Phi(x^k(\omega))\|^b)^{1/q}$ for all $k \geq K$. We first consider the case $qa > b$. It follows from (P.3′) that

$$\|x^{u_t}(\omega) - x^{\ell_t}(\omega)\| \leq B\sum_{k=\ell_t}^{u_t-1}\mu_k^{p_2}(1 + \|\Phi(x^k(\omega))\|^b)^{\frac{1}{q}} + \left\|\sum_{k=\ell_t}^{u_t-1}\mu_k^{p_1}A_k(\omega)\right\|$$

$$\leq B\sum_{k=\ell_t}^{u_t-1}\mu_k + B\sum_{k=\ell_t}^{u_t-1}\mu_k\|\Phi(x^k(\omega))\|^{\frac{b}{q}} + \left\|\sum_{k=\ell_t}^{u_t-1}\mu_k^{p_1}A_k(\omega)\right\|$$

$$\leq B\beta_t + B\left(\sum_{k=\ell_t}^{u_t-1}\mu_k\right)^{1-\frac{b}{qa}}\left(\sum_{k=\ell_t}^{u_t-1}\mu_k\|\Phi(x^k(\omega))\|^a\right)^{\frac{b}{qa}}$$

$$+ \|M_{u_t}(\omega) - M_{\ell_t}(\omega)\|$$

$$\leq B\beta_t + B\beta_t^{1-\frac{b}{qa}}\left(\sum_{k=0}^{\infty}\mu_k\|\mathbf{\Phi}(\boldsymbol{x}^k(\omega))\|^a\right)^{\frac{b}{qa}} + \|\boldsymbol{M}_{u_t}(\omega) - \boldsymbol{M}_{\ell_t}(\omega)\|$$

for all $t$ sufficiently large, where we used the subadditivity of $x \mapsto x^{\frac{1}{q}}$, $p_2 \geq 1$, and $\mu_k \to 0$ in the second inequality, and Hölder's inequality in the third inequality. In the case $qa = b$, let us introduce $\boldsymbol{E}_k(\omega) := \sum_{i=0}^{k-1}\mu_k\|\mathbf{\Phi}(\boldsymbol{x}^k(\omega))\|^a$. Then, it follows

$$\|\boldsymbol{x}^{u_t}(\omega) - \boldsymbol{x}^{\ell_t}(\omega)\| \leq B\beta_t + B[\boldsymbol{E}_{u_t}(\omega) - \boldsymbol{E}_{\ell_t}(\omega)] + \|\boldsymbol{M}_{u_t}(\omega) - \boldsymbol{M}_{\ell_t}(\omega)\|$$

for all $t$ sufficiently large. Due to $\beta_t \to 0$, $\boldsymbol{E}_k(\omega) \to \boldsymbol{E}(\omega) := \sum_{k=0}^{\infty}\mu_k\|\mathbf{\Phi}(\boldsymbol{x}^k(\omega))\|^a$, and $\boldsymbol{M}_k(\omega) \to \boldsymbol{M}(\omega)$, we can infer $\|\boldsymbol{x}^{u_t}(\omega) - \boldsymbol{x}^{\ell_t}(\omega)\| \to 0$ in both cases. As before, the Lipschitz continuity of $\mathbf{\Phi}$ in (P.1) yields the contradiction

$$\delta \leq \|\mathbf{\Phi}(\boldsymbol{x}^{\ell_t}(\omega))\| - \|\mathbf{\Phi}(\boldsymbol{x}^{u_t}(\omega))\| \leq \|\mathbf{\Phi}(\boldsymbol{x}^{u_t}(\omega)) - \mathbf{\Phi}(\boldsymbol{x}^{\ell_t}(\omega))\| \leq \mathsf{L}_{\mathbf{\Phi}}\|\boldsymbol{x}^{u_t}(\omega) - \boldsymbol{x}^{\ell}(\omega)\| \to 0.$$

Thus, for all $\omega \in \mathcal{M}$, we have $\lim_{k\to\infty}\|\mathbf{\Phi}(\boldsymbol{x}^k(\omega))\| = 0$. Since the event $\mathcal{M}$ occurs with probability 1, this concludes the proof of item (ii). $\blacksquare$

## B  The supermartingale convergence theorem

The following well-known and celebrated convergence theorem for supermartingale-type stochastic processes is due to Robbins and Siegmund [37].

**Theorem B.1** (Supermartingale Convergence Theorem). *Let $\{y_k\}_{k\geq 0}$, $\{p_k\}_{k\geq 0}$, and $\{q_k\}_{k\geq 0}$ be sequences of nonnegative integrable random variables adapted to a filtration $\{\mathcal{U}_k\}_{k\geq 0}$. Furthermore, let $\{\beta_k\}_{k\geq 0} \subseteq \mathbb{R}_+$ be given with $\sum_{k=0}^{\infty}\beta_k < \infty$ and assume that we have*

$$\mathbb{E}[y_{k+1} \mid \mathcal{U}_k] \leq (1 + \beta_k)y_k - p_k + q_k$$

*for all $k$ and $\sum_{k=0}^{\infty}q_k < \infty$ almost surely. Then, it holds that*

(a) *If $\sum_{k=0}^{\infty}\mathbb{E}[q_k] < \infty$, then the sequence $\{\mathbb{E}[y_k]\}_{k\geq 0}$ converges to a finite number $\overline{y}$ and we have $\sum_{k=0}^{\infty}\mathbb{E}[p_k] < \infty$.*

(b) *$\{y_k\}_{k\geq 0}$ almost surely converges to a nonnegative finite random variable $y$ and it follows $\sum_{k=0}^{\infty}p_k < \infty$ almost surely.*

## C  Proofs for Subsection 3.1

### C.1  Derivation of (6)

Using the Lipschitz continuity of $\nabla f$ and the descent lemma, we obtain

$$f(\boldsymbol{x}^{k+1}) \leq f(\boldsymbol{x}^k) + \langle \nabla f(\boldsymbol{x}^k), \boldsymbol{x}^{k+1} - \boldsymbol{x}^k \rangle + \frac{\mathsf{L}}{2}\|\boldsymbol{x}^{k+1} - \boldsymbol{x}^k\|^2$$

$$= f(\boldsymbol{x}^k) + \alpha_k\langle \nabla f(\boldsymbol{x}^k), \nabla f(\boldsymbol{x}^k) - \boldsymbol{g}^k \rangle - \alpha_k\|\nabla f(\boldsymbol{x}^k)\|^2 + \frac{\mathsf{L}\alpha_k^2}{2}\|\boldsymbol{g}^k\|^2$$

$$= f(\boldsymbol{x}^k) - \alpha_k\left(1 - \frac{\mathsf{L}\alpha_k}{2}\right)\|\nabla f(\boldsymbol{x}^k)\|^2 + \alpha_k(1 - \mathsf{L}\alpha_k)\langle \nabla f(\boldsymbol{x}^k), \nabla f(\boldsymbol{x}^k) - \boldsymbol{g}^k \rangle$$

$$+ \frac{\mathsf{L}\alpha_k^2}{2}\|\nabla f(\boldsymbol{x}^k) - \boldsymbol{g}^k\|^2.$$

Taking conditional expectation gives (6).

## D  Results and proofs for Subsection 3.2

We now provide more detailed algorithmic procedure and motivations for RR. First, let us define the set of all possible permutations of $\{1, 2, \ldots, N\}$ as

$$\Lambda := \{\sigma : \sigma \text{ is a permutation of } \{1, 2, \ldots, N\}\}.$$

At each iteration $k$, a permutation $\sigma^{k+1}$ is generated according to an i.i.d. uniform distribution over $\Lambda$. Then, RR updates $\boldsymbol{x}^k$ to $\boldsymbol{x}^{k+1}$ through $N$ consecutive gradient descent-type steps by using the

components $\{f(\cdot, \sigma_1^{k+1}), \dots, f(\cdot, \sigma_N^{k+1})\}$ sequentially, where $\sigma_i^{k+1}$ represents the $i$-th element of $\sigma^{k+1}$. In each step, only one component $f(\cdot, \sigma_i^{k+1})$ is selected for updating. To be more specific, this method starts with $\tilde{\boldsymbol{x}}_0^k = \boldsymbol{x}^k$ and then uses $f(\cdot, \sigma_i^{k+1})$ to update $\tilde{\boldsymbol{x}}_i^k$ as

$$\tilde{\boldsymbol{x}}_i^k = \tilde{\boldsymbol{x}}_{i-1}^k - \alpha_k \nabla f(\tilde{\boldsymbol{x}}_{i-1}^k, \sigma_i^{k+1})$$

for $i = 1, 2, \dots, N$, resulting in $\boldsymbol{x}^{k+1} = \tilde{\boldsymbol{x}}_N^k$.

RR is used in a vast variety of engineering fields. Most notably, RR is extensively applied in practice for training deep neural networks; see, e.g., [20, 30, 32] and the references therein.

### D.1 A bound on the step length

We need the following lemma to for our later analysis.

**Lemma D.1.** *We have the following estimate:*

$$\|\boldsymbol{x}^{k+1} - \boldsymbol{x}^k\|^q \leq \bar{\mathsf{A}} \mathsf{G}(\boldsymbol{x}^0)^{\frac{q}{2}} \alpha_k^q \qquad (20)$$

*for all $k$ sufficiently large and some positive constants $\bar{\mathsf{A}}$ and $\mathsf{G}(\boldsymbol{x}^0)^{\frac{q}{2}}$.*

*Proof.* Using $|\sum_{i=1}^N a_i|^q \leq N^{q-1} \sum_{i=1}^N |a_i|^q$, (7), and $\nabla f(\boldsymbol{x}^k) = \frac{1}{N} \sum_{i=1}^N \nabla f(\boldsymbol{x}^k, \sigma_i^{k+1})$, we have

$$\|\boldsymbol{x}^{k+1} - \boldsymbol{x}^k\|^q = \alpha_k^q \left\| \sum_{i=1}^N \nabla f(\tilde{\boldsymbol{x}}_{i-1}^k, \sigma_i^{k+1}) \right\|^q$$

$$\leq 2^{q-1} N^q \alpha_k^q \|\nabla f(\boldsymbol{x}^k)\|^q + 2^{q-1} \alpha_k^q \left\| \sum_{i=1}^N [\nabla f(\tilde{\boldsymbol{x}}_{i-1}^k, \sigma_i^{k+1}) - \nabla f(\boldsymbol{x}^k, \sigma_i^{k+1})] \right\|^q$$

$$\leq 2^{q-1} N^q \alpha_k^q \|\nabla f(\boldsymbol{x}^k)\|^q + (2N)^{q-1} \mathsf{L}^q \alpha_k^q \cdot \sum_{i=1}^N \|\tilde{\boldsymbol{x}}_{i-1}^k - \boldsymbol{x}^k\|^q.$$

Setting $V_k := \sum_{i=1}^N \|\tilde{\boldsymbol{x}}_{i-1}^k - \boldsymbol{x}^k\|^q$, we recursively obtain

$$V_k = \alpha_k^q \sum_{i=2}^N \left\| \sum_{j=1}^{i-1} \nabla f(\tilde{\boldsymbol{x}}_{j-1}^k, \sigma_j^{k+1}) \right\|^q$$

$$\leq 2^{q-1} \alpha_k^q \sum_{i=2}^N \left[ \left\| \sum_{j=1}^{i-1} [\nabla f(\tilde{\boldsymbol{x}}_{j-1}^k, \sigma_j^{k+1}) - \nabla f(\boldsymbol{x}^k, \sigma_j^{k+1})] \right\|^q + \left\| \sum_{j=1}^{i-1} \nabla f(\boldsymbol{x}^k, \sigma_j^{k+1}) \right\|^q \right]$$

$$\leq 2^{q-1} \alpha_k^q \sum_{i=2}^N \left[ \mathsf{L}^q (i-1)^{q-1} \sum_{j=1}^{i-1} \|\tilde{\boldsymbol{x}}_{j-1}^k - \boldsymbol{x}^k\|^q + \left[ (i-1) \sum_{j=1}^{i-1} \|\nabla f(\boldsymbol{x}^k, \sigma_j^{k+1})\|^2 \right]^{\frac{q}{2}} \right]$$

$$\leq 2^{q-1} \mathsf{L}^q \alpha_k^q \left( \sum_{i=1}^{N-1} i^{q-1} \right) V_k + 2^{\frac{3q}{2}-1} \mathsf{L}^{\frac{q}{2}} \alpha_k^q \sum_{i=2}^N (i-1)^{\frac{q}{2}} \left[ \sum_{j=1}^{i-1} (f(\boldsymbol{x}^k, \sigma_j^{k+1}) - \bar{f}) \right]^{\frac{q}{2}}$$

$$\leq 2^{q-1} \mathsf{L}^q \left[ \frac{N^q}{q} \right] \alpha_k^q V_k + 2^{\frac{3q-2}{2}} (\mathsf{L}N)^{\frac{q}{2}} \left[ \frac{2N^{\frac{q}{2}+1}}{q+2} \right] \alpha_k^q (f(\boldsymbol{x}^k) - \bar{f})^{\frac{q}{2}}$$

where we applied the estimate $\|\nabla f(\boldsymbol{x}, i)\|^2 \leq 2\mathsf{L}(f(\boldsymbol{x}, i) - \bar{f})$ (see also (24) for comparison). Clearly, this establishes $V_k = \mathcal{O}(\alpha_k^q (f(\boldsymbol{x}^k) - \bar{f})^{\frac{q}{2}})$. Furthermore, following the proof of [25, Lemma 3.2], we have $f(\boldsymbol{x}^k) - \bar{f} \leq \mathsf{G}(\boldsymbol{x}^0)$ where $\mathsf{G}(\boldsymbol{x}^0) = (f(\boldsymbol{x}^0) - \bar{f}) \exp(2\mathsf{L}^3 N^3 \sum_{j=0}^{\infty} \alpha_j^3)$. Hence, using $\|\nabla f(\boldsymbol{x})\|^2 \leq 2\mathsf{L}(f(\boldsymbol{x}) - \bar{f})$ and (B.2), there exists a constant $\bar{\mathsf{A}}$ such that (20) holds for all sufficiently large $k$. $\qquad\square$

### D.2 Proof of Corollary 3.2

We derive the convergence of RR below by setting $\boldsymbol{\Phi} \equiv \nabla f$ and $\mu_k \equiv \alpha_k$.

**Phase I: Verifying (P.1)–(P.2).** (B.1) verifies condition (P.1) with $\mathsf{L}_{\Phi} = \mathsf{L}$. Towards verifying (P.2), note that under the above assumptions for RR, [25, Lemma 3.1] establishes the recursion:

$$f(\boldsymbol{x}^{k+1}) - \bar{f} \leq (1 + 2\mathsf{L}^3 N^3 \alpha_k^3)[f(\boldsymbol{x}^k) - \bar{f}] - \frac{N\alpha_k}{2} \|\nabla f(\boldsymbol{x}^k)\|^2 - \frac{1 - \mathsf{L}N\alpha_k}{2N\alpha_k} \|\boldsymbol{x}^{k+1} - \boldsymbol{x}^k\|^2 \quad (21)$$

for all $k$ as long as $\alpha_k < 1/(\sqrt{2}\mathsf{L}N)$. (This always holds for large enough $k$ as $\alpha_k \to 0$). Taking total expectation and applying Theorem B.1 provides $\mathbb{E}[f(\boldsymbol{x}^k) - \bar{f}] \to \mathsf{F}$ and $\sum_{k=0}^{\infty} \alpha_k \mathbb{E}[\|\nabla f(\boldsymbol{x}^k)\|^2] < \infty$. This verifies condition (P.2) with $a = 2$.

**Phase II: Verifying (P.3)–(P.4) for showing expected convergence.** We can infer from Lemma D.1 that (P.3) holds with arbitrary $q \geq 1$ and $q \in \mathbb{N}$, $\mathsf{A} = \bar{\mathsf{A}}\mathsf{G}(\boldsymbol{x}^0)^{\frac{q}{2}}$, $p_1 = q$, and $\mathsf{B} = 0$. (P.4) can be easily verified by these parameter choices and (B.2). Thus, applying Theorem 2.1 gives $\mathbb{E}[\|\nabla f(\boldsymbol{x}^k)\|] \to 0$.

**Phase III: Verifying (P.3′)–(P.4′) for showing almost sure convergence.** According to the update (7), we can let $\boldsymbol{A}_k \equiv 0$, $p_2 = 1$, $\boldsymbol{B}_k = \sum_{i=1}^{N} \nabla f(\tilde{\boldsymbol{x}}_{i-1}^k, \sigma_i^{k+1})$, $q = 2$, and $b = 2$, in (P.3′). We have

$$\limsup_{k \to \infty} \frac{\|\boldsymbol{B}_k\|^2}{1 + \|\nabla f(\boldsymbol{x}^k)\|^2} \leq \limsup_{k \to \infty} \frac{2\|\boldsymbol{B}_k - N\nabla f(\boldsymbol{x}^k)\|^2 + 2N^2\|\nabla f(\boldsymbol{x}^k)\|^2}{1 + \|\nabla f(\boldsymbol{x}^k)\|^2} < \infty,$$

since $\|\boldsymbol{B}_k - N\nabla f(\boldsymbol{x}^k)\|^2 \leq \mathcal{O}(\alpha_k^2)$ — as shown in Appendix D.1 — which converges to 0 as $k \to \infty$. Condition (P.3′) is verified. Assumption (B.2) and the previous parameter choices verify condition (P.4′). Hence, we can apply Theorem 2.1 to derive $\|\nabla f(\boldsymbol{x}^k)\| \to 0$ almost surely.

Summarizing the above results yields Corollary 3.2.

# E    Results and proofs for Subsection 3.3

## E.1    Weakly convex functions

A function $h : \mathbb{R}^n \to (-\infty, \infty]$ is said to be $\tau$-weakly convex if

$$h + \frac{\tau}{2}\|\cdot\|^2$$

is convex (on its effective domain $\operatorname{dom} h = \{\boldsymbol{x} \in \mathbb{R}^n : h(\boldsymbol{x}) < \infty\}$). The class of weakly convex functions covers many important nonsmooth nonconvex problems. For instance, any function that has the composite form

$$h(\boldsymbol{x}) = r(c(\boldsymbol{x}))$$

with $r$ being convex Lipschitz continuous and the Jacobian of $c$ being Lipschitz continuous is weakly convex. We refer to [11] for more discussions.

If $h$ is $\tau$-weakly convex, proper, and lower semicontinuous, then the proximity operator $\operatorname{prox}_{\alpha h} : \mathbb{R}^n \to \mathbb{R}^n$, given by

$$\operatorname{prox}_{\alpha h}(\boldsymbol{x}) := \operatorname*{argmin}_{\boldsymbol{y} \in \mathbb{R}^n} \ h(\boldsymbol{y}) + \frac{1}{2\alpha}\|\boldsymbol{x} - \boldsymbol{y}\|^2,$$

is a well-defined function for all $\alpha \in (0, \tau^{-1})$ and Lipschitz continuous with constant $(1 - \alpha\tau)^{-1}$, see, e.g., [39, Proposition 12.19] or [21, Proposition 3.3]. The $\tau$-weak convexity is equivalent to

$$h(\boldsymbol{y}) \geq h(\boldsymbol{x}) + \langle \boldsymbol{s}, \boldsymbol{y} - \boldsymbol{x} \rangle - \frac{\tau}{2}\|\boldsymbol{x} - \boldsymbol{y}\|^2, \quad \forall \boldsymbol{x}, \boldsymbol{y}, \ \forall \boldsymbol{s} \in \partial h(\boldsymbol{x}) \tag{22}$$

see, e.g., [44, 11].

## E.2    Examples of weakly convex and Lipschitz continuous regularizers

In this section, we discuss several common nonconvex regularizers that can be shown to be weakly convex and Lipschitz continuous. The functions presented here have been mentioned in Remark 3.3.

The minimax concave penalty (MCP), introduced in [45], is the parametrized function $\varphi_{\lambda,\theta} : \mathbb{R} \to \mathbb{R}_+$ given by

$$\varphi_{\lambda,\theta}(x) := \begin{cases} \lambda|x| - \frac{x^2}{2\theta} & \text{if } |x| \leq \theta\lambda, \\ \frac{\theta\lambda^2}{2} & \text{otherwise,} \end{cases}$$

where $\lambda, \theta > 0$ are two positive parameters. This function is $\theta^{-1}$-weakly convex and smooth for $x \neq 0$. Discussing the subdifferential $\partial\varphi_{\lambda,\theta}(x)$ and using [39, Theorem 9.13], it can be shown that $\varphi_{\lambda,\theta}$ is Lipschitz continuous on $\mathbb{R}$ with modulus $\lambda$.

The smoothly clipped absolute deviation (SCAD) [15] is defined by

$$
\varphi_{\lambda,\theta}(x) := \begin{cases} \lambda|x| & \text{if } |x| \le \lambda, \\ \frac{-x^2 + 2\theta\lambda|x| - \lambda^2}{2(\theta-1)} & \text{if } \lambda < |x| \le \theta\lambda, \\ \frac{(\theta+1)\lambda^2}{2} & \text{if } |x| > \theta\lambda, \end{cases}
$$

where $\lambda > 0$ and $\theta > 2$ are given parameters. The SCAD function is $(\theta-1)^{-1}$-weakly convex and Lipschitz continuous with modulus $\lambda$.

The student-t loss function is given by $\varphi_\theta(x) := \frac{\theta^2}{2}\log(1 + \theta^{-2}x^2)$ for some $\theta \neq 0$. The first- and second-order derivative of $\varphi_\theta$ can be calculated as follows:

$$
\varphi'_\theta(x) = \frac{\theta^2 x}{\theta^2 + x^2} \quad \text{and} \quad \varphi''_\theta(x) = \frac{\theta^2(\theta^2 - x^2)}{(\theta^2 + x^2)^2}.
$$

Some simple computations then show that $\varphi_\theta$ is $\frac{1}{8}$-weakly convex and Lipschitz continuous with modulus $|\theta|/2$. Additional examples can be found in [4].

### E.3 Equivalent stationarity measures

We now first show that the two stationarity measures $x \mapsto \|F^1_{\text{nat}}(x)\|$ and $x \mapsto \|\nabla\text{env}_{\theta\psi}(x)\|$ are equivalent. This can be used to verify Corollary 3.6.

**Lemma E.1.** *Suppose that the conditions (C.1) and (C.2) are satisfied and let $\theta \in (0, [3L + \tau]^{-1})$ and $x \in \mathbb{R}^n$ be given. Then, we have*

$$
(1 - [3L + \tau]\theta)\gamma\theta^{-2}\|F_{\text{nat}}(x)\| \le \|\nabla\text{env}_{\theta\psi}(x)\| \le (1 + [L - \tau]\theta)(\gamma + \tau)\theta^{-2}\|F_{\text{nat}}(x)\|,
$$

*where $\gamma = \theta/(1 - [L + \tau]\theta)$ and $F_{\text{nat}} := F^1_{\text{nat}}$.*

*Proof.* For every fixed $x \in \mathbb{R}^n$, let us define $\psi_x(y) := \psi(y) + \frac{L+\tau}{2}\|x - y\|^2$ and $\varphi_x(y) := \varphi(y) + \frac{\tau}{2}\|x - y\|^2$. Then, setting $\gamma = \theta/(1 - [L + \tau]\theta)$, we may write

$$
\text{prox}_{\theta\psi}(x) = \text{prox}_{\gamma\psi_x}(x) \quad \text{and} \quad \nabla\text{env}_{\theta\psi}(x) = \theta^{-1}\gamma\nabla\text{env}_{\gamma\psi_x}(x).
$$

Applying [13, Theorem 3.5] with $G \equiv \partial\varphi_x$, $\Phi \equiv \partial\psi_x$, $F \equiv \nabla f + L(\cdot - x)$, $t \equiv \gamma$, and $\beta \equiv 2L$, we can establish the following estimates for all $x \in \mathbb{R}^n$:

$$
(1 - 2L\gamma)\theta^{-1}\|x - \text{prox}_{\gamma\varphi_x}(x - \gamma F(x))\| \le \|\nabla\text{env}_{\theta\psi}(x)\|
$$

and $\|\nabla\text{env}_{\theta\psi}(x)\| \le (1 + 2L\gamma)\theta^{-1}\|x - \text{prox}_{\gamma\varphi_x}(x - \gamma F(x))\|$. Moreover, it holds that

$$
\begin{aligned}
\text{prox}_{\gamma\varphi_x}(x - \gamma F(x)) &= \text{prox}_{\gamma\varphi_x}(x - \gamma\nabla f(x)) \\
&= \underset{y \in \mathbb{R}^n}{\arg\min}\ \varphi(y) + \langle\nabla f(x), y - x\rangle + \frac{1}{2}\left[\frac{1}{\gamma} + \tau\right]\|y - x\|^2 \\
&= \text{prox}_{\frac{\gamma}{\gamma+\tau}\varphi}\left(x - \frac{\gamma}{\gamma+\tau}\nabla f(x)\right)
\end{aligned}
$$

In addition, as shown in [31, Lemma 2], the functions $\delta \mapsto \|F^{1/\delta}_{\text{nat}}(x)\|$ and $\delta \mapsto \|F^{1/\delta}_{\text{nat}}(x)\|/\delta$ are decreasing and increasing in $\delta$, respectively. Hence, we can infer

$$
\min\{1, \lambda_2/\lambda_1\}\|F^{\lambda_2}_{\text{nat}}(x)\| \le \|F^{\lambda_1}_{\text{nat}}(x)\| \le \max\{1, \lambda_2/\lambda_1\}\|F^{\lambda_2}_{\text{nat}}(x)\| \tag{23}
$$

for all $\lambda_1, \lambda_2 > 0$ and $x \in \mathbb{R}^n$. Choosing $\lambda_1 = \gamma/(\gamma + \tau)$ and $\lambda_2 = 1$ and noticing $1 - 2L\gamma = (1 - [3L + \tau]\theta)\theta^{-1} > 0$, we can conclude the proof of Lemma E.1. $\square$

### E.4 Proof of Lemma 3.4

Let us again note that the $\tau$-weak convexity of $\varphi$ — as stated in (C.2) — implies Lipschitz continuity of the proximity operator $\text{prox}_{\theta\varphi}$ with modulus $(1 - \theta\tau)^{-1}$. In the following results, we first bound the term $\|x^{k+1} - \bar{x}^k\|^2 = \|x^{k+1} - \text{prox}_{\theta\psi}(x^k)\|^2$.

**Lemma E.2.** *Suppose that* (C.1) *and* (C.2) *are satisfied and let* $\theta \in (0, [\mathsf{L} + \tau]^{-1})$, $\alpha > 0$, $\boldsymbol{x} \in \mathbb{R}^n$, *and* $\bar{\boldsymbol{x}} = \mathrm{prox}_{\theta\psi}(\boldsymbol{x})$ *be given. Then, it holds that*

$$\bar{\boldsymbol{x}} = \mathrm{prox}_{\alpha\varphi}(\bar{\boldsymbol{x}} - \alpha\nabla f(\bar{\boldsymbol{x}}) - \alpha\theta^{-1}[\bar{\boldsymbol{x}} - \boldsymbol{x}]).$$

*Proof.* This is Lemma 3.2 in [11]. □

Our second result is analogous to Lemma 3.3 in [11].

**Lemma E.3.** *Suppose that the conditions* (C.1), (C.2), *and* (C.4) *are satisfied and let* $\theta \in (0, [\mathsf{L}+\tau]^{-1})$ *be given. Defining* $\bar{\boldsymbol{x}}^k = \mathrm{prox}_{\theta\psi}(\boldsymbol{x}^k)$, $k \in \mathbb{N}$, *it holds that*

$$\mathbb{E}[\|\boldsymbol{x}^{k+1} - \bar{\boldsymbol{x}}^k\|^2 \mid \mathcal{F}_k] \le (1 - \alpha_k\theta_k)^2\|\boldsymbol{x}^k - \bar{\boldsymbol{x}}^k\|^2 + \frac{\alpha_k^2\sigma_k^2}{(1 - \alpha_k\tau)^2} \quad \text{almost surely} \quad \forall\, k \in \mathbb{N},$$

*where* $\theta_k := (\theta^{-1} - [\mathsf{L} + \tau])/(1 - \alpha_k\tau)$ *and* $\sigma_k^2 := \mathbb{E}[\|\boldsymbol{g}^k - \nabla f(\boldsymbol{x}^k)\|^2 \mid \mathcal{F}_k]$.

*Proof.* Invoking Lemma E.2 and the Lipschitz continuity of $\mathrm{prox}_{\alpha_k\varphi}$ and $\nabla f$, it follows

$$(1 - \alpha_k\tau)^2\|\boldsymbol{x}^{k+1} - \bar{\boldsymbol{x}}^k\|^2$$
$$= (1 - \alpha_k\tau)^2\|\mathrm{prox}_{\alpha_k\varphi}(\boldsymbol{x}^k - \alpha_k\boldsymbol{g}^k) - \mathrm{prox}_{\alpha_k\varphi}(\bar{\boldsymbol{x}}^k - \alpha_k\nabla f(\bar{\boldsymbol{x}}^k) - \alpha_k\theta^{-1}[\bar{\boldsymbol{x}}^k - \boldsymbol{x}^k])\|^2$$
$$\le \|(1 - \alpha_k\theta^{-1})[\bar{\boldsymbol{x}}^k - \boldsymbol{x}^k] + \alpha_k[\boldsymbol{g}^k - \nabla f(\bar{\boldsymbol{x}}^k)]\|^2$$
$$= (1 - \alpha_k\theta^{-1})^2\|\bar{\boldsymbol{x}}^k - \boldsymbol{x}^k\|^2 + 2\alpha_k(1 - \alpha_k\theta^{-1})\langle\bar{\boldsymbol{x}}^k - \boldsymbol{x}^k, \boldsymbol{g}^k - \nabla f(\bar{\boldsymbol{x}}^k)\rangle$$
$$\quad + \alpha_k^2[\|\boldsymbol{g}^k - \nabla f(\boldsymbol{x}^k)\|^2 + 2\langle\boldsymbol{g}^k - \nabla f(\boldsymbol{x}^k), \nabla f(\boldsymbol{x}^k) - \nabla f(\bar{\boldsymbol{x}}^k)\rangle + \|\nabla f(\boldsymbol{x}^k) - \nabla f(\bar{\boldsymbol{x}}^k)\|^2]$$
$$\le (1 - \alpha_k[\theta^{-1} - \mathsf{L}])^2\|\bar{\boldsymbol{x}}^k - \boldsymbol{x}^k\|^2 + 2\alpha_k(1 - \alpha_k\theta^{-1})\langle\bar{\boldsymbol{x}}^k - \boldsymbol{x}^k, \boldsymbol{g}^k - \nabla f(\boldsymbol{x}^k)\rangle$$
$$\quad + 2\alpha_k^2\langle\boldsymbol{g}^k - \nabla f(\boldsymbol{x}^k), \nabla f(\boldsymbol{x}^k) - \nabla f(\bar{\boldsymbol{x}}^k)\rangle + \alpha_k^2\|\boldsymbol{g}^k - \nabla f(\boldsymbol{x}^k)\|^2.$$

Taking conditional expectation, using (C.4) and $\boldsymbol{x}^k, \bar{\boldsymbol{x}}^k \in \mathcal{F}_k$, we obtain

$$\mathbb{E}[\|\boldsymbol{x}^{k+1} - \bar{\boldsymbol{x}}^k\|^2 \mid \mathcal{F}_k] \le (1 - \alpha_k\theta_k)^2\|\boldsymbol{x}^k - \bar{\boldsymbol{x}}^k\|^2 + \frac{\alpha_k^2\sigma_k^2}{(1 - \alpha_k\tau)^2}$$

almost surely, where $\theta_k = (\theta^{-1} - [\mathsf{L} + \tau])/(1 - \alpha_k\tau)$. □

By (C.4), the stochastic error term $\sigma_k^2$ is bounded by $\mathsf{C}[f(\boldsymbol{x}^k) - \bar{f}] + \mathsf{D}$ almost surely. In order to proceed, we need to link the function values "$f(\boldsymbol{x}^k) - \bar{f}$" and "$\mathrm{env}_{\theta\psi}(\boldsymbol{x}^k) - \bar{\psi}$" where $\bar{\psi} = \bar{f} + \bar{\varphi}$. The following lemma precisely establishes such a connection under assumption (C.3).

**Lemma E.4.** *Suppose that the conditions* (C.1)–(C.3) *are satisfied and let* $\theta \in (0, [\frac{4}{3}\mathsf{L} + \tau]^{-1})$ *be given. Then, it holds that*

$$f(\boldsymbol{x}) - \bar{f} \le 2[\mathrm{env}_{\theta\psi}(\boldsymbol{x}) - \bar{\psi}] + \mathsf{L}_\varphi^2\theta \quad \forall\, \boldsymbol{x} \in \mathrm{dom}\,\varphi.$$

*Proof.* Notice that the Lipschitz continuity of $\nabla f$ and assumption (C.2) imply $f(\boldsymbol{x} - \mathsf{L}^{-1}\nabla f(\boldsymbol{x})) - f(\boldsymbol{x}) \le -\frac{1}{2\mathsf{L}}\|\nabla f(\boldsymbol{x})\|^2$ and

$$\|\nabla f(\boldsymbol{x})\|^2 \le 2\mathsf{L}[f(\boldsymbol{x}) - \bar{f}] = 2\mathsf{L}[\psi(\boldsymbol{x}) - \varphi(\boldsymbol{x}) - \bar{f}] \le 2\mathsf{L}[\psi(\boldsymbol{x}) - \bar{\psi}] \quad (24)$$

for all $\boldsymbol{x} \in \mathrm{dom}\,\varphi$. Using (C.3), the Lipschitz continuity of $\nabla f$, Young's inequality (twice), and $\theta < \frac{3}{4}\mathsf{L}^{-1}$ and setting $\bar{\boldsymbol{x}} = \mathrm{prox}_{\theta\psi}(\boldsymbol{x}) \in \mathrm{dom}\,\varphi$, this yields

$$\psi(\boldsymbol{x}) - \bar{\psi} \le \mathrm{env}_{\theta\psi}(\boldsymbol{x}) - \bar{\psi} + \langle\nabla f(\bar{\boldsymbol{x}}), \boldsymbol{x} - \bar{\boldsymbol{x}}\rangle + \frac{1}{2}\left[\mathsf{L} - \theta^{-1}\right]\|\boldsymbol{x} - \bar{\boldsymbol{x}}\|^2 + \varphi(\boldsymbol{x}) - \varphi(\bar{\boldsymbol{x}})$$

$$\le \mathrm{env}_{\theta\psi}(\boldsymbol{x}) - \bar{\psi} + \frac{1}{2\mathsf{L}}\|\nabla f(\bar{\boldsymbol{x}})\|^2 + \left[\mathsf{L} - \frac{1}{2\theta}\right]\|\boldsymbol{x} - \bar{\boldsymbol{x}}\|^2 + \frac{1}{4\theta}\|\boldsymbol{x} - \bar{\boldsymbol{x}}\|^2 + \mathsf{L}_\varphi^2\theta$$

$$\le \mathrm{env}_{\theta\psi}(\boldsymbol{x}) - \bar{\psi} + [\psi(\bar{\boldsymbol{x}}) - \bar{\psi}] + \frac{1}{2\theta}\|\boldsymbol{x} - \bar{\boldsymbol{x}}\|^2 + \mathsf{L}_\varphi^2\theta = 2[\mathrm{env}_{\theta\psi}(\boldsymbol{x}) - \bar{\psi}] + \mathsf{L}_\varphi^2\theta.$$

This finishes the proof of Lemma E.4. □

We now verify Lemma 3.4. Choosing $\theta \in (0, [3\mathsf{L} + \tau]^{-1})$ ensures that Lemma E.1 and Lemma E.4 are applicable. Let $k \in \mathbb{N}$ be given with $\alpha_k \leq \min\{\frac{1}{2\tau}, \frac{1}{2(\theta^{-1} - [\mathsf{L} + \tau])}\}$. This implies $1 - \alpha_k \tau > \frac{1}{2}$,

$$2\mathsf{L} \leq \theta^{-1} - [\mathsf{L} + \tau] \leq \theta_k \leq 2(\theta^{-1} - [\mathsf{L} + \tau]),$$

and $2 - \theta_k \alpha_k \geq 1$. Utilizing the definition of the Moreau envelope, we can further infer

$$
\begin{aligned}
\mathrm{env}_{\theta\psi}(\boldsymbol{x}^{k+1}) &= \min_{\boldsymbol{y} \in \mathbb{R}^n} \ \psi(\boldsymbol{y}) + \frac{1}{2\theta}\|\boldsymbol{y} - \boldsymbol{x}^{k+1}\|^2 \\
&\leq \mathrm{env}_{\theta\psi}(\boldsymbol{x}^k) + \frac{1}{2\theta}\left[\|\bar{\boldsymbol{x}}^k - \boldsymbol{x}^{k+1}\|^2 - \|\bar{\boldsymbol{x}}^k - \boldsymbol{x}^k\|^2\right]. \quad (25)
\end{aligned}
$$

Taking conditional expectation in (25) and applying Lemma E.3, (C.4), and Lemma E.4, we obtain

$$
\begin{aligned}
\mathbb{E}[\mathrm{env}_{\theta\psi}(\boldsymbol{x}^{k+1}) - \bar{\psi} \mid \mathcal{F}_k] &\leq [\mathrm{env}_{\theta\psi}(\boldsymbol{x}^k) - \bar{\psi}] + \frac{1}{2\theta}\left[\mathbb{E}[\|\boldsymbol{x}^{k+1} - \bar{\boldsymbol{x}}^k\|^2 \mid \mathcal{F}_k] - \|\bar{\boldsymbol{x}}^k - \boldsymbol{x}^k\|^2\right], \\
&\leq [\mathrm{env}_{\theta\psi}(\boldsymbol{x}^k) - \bar{\psi}] - \frac{2 - \theta_k \alpha_k}{2\theta}\theta_k \alpha_k \|\bar{\boldsymbol{x}}^k - \boldsymbol{x}^k\|^2 + \frac{\alpha_k^2 \sigma_k^2}{2\theta(1 - \alpha_k \tau)^2} \\
&\leq \left[1 + \frac{4\mathsf{C}\alpha_k^2}{\theta}\right][\mathrm{env}_{\theta\psi}(\boldsymbol{x}^k) - \bar{\psi}] - \mathsf{L}\theta\alpha_k \|\nabla\mathrm{env}_{\theta\psi}(\boldsymbol{x}^k)\|^2 + 2\alpha_k^2\left[\mathsf{C}\mathsf{L}_\varphi^2 + \frac{\mathsf{D}}{\theta}\right],
\end{aligned}
$$

almost surely, where we used $\boldsymbol{x}^k - \bar{\boldsymbol{x}}^k = \theta\nabla\mathrm{env}_{\theta\psi}(\boldsymbol{x}^k)$. This finishes the proof of Lemma 3.4.

### E.5  Proof of Lemma 3.5

We first establish an upper bound for the natural residual that is analogous to [12, Lemma A.1].

**Lemma E.5.** *Suppose that the conditions (C.2) and (C.3) are satisfied. Let $\boldsymbol{x} \in \mathrm{dom}\,\varphi$, $\boldsymbol{v} \in \mathbb{R}^n$, and $\alpha \in (0, \frac{1}{2}\tau^{-1})$ be given and set $F_v^\alpha(\boldsymbol{x}) := \boldsymbol{x} - \mathrm{prox}_{\alpha\varphi}(\boldsymbol{x} - \alpha\boldsymbol{v})$. Then, it holds that*

$$\|F_v^\alpha(\boldsymbol{x})\|^2 \leq 4\mathsf{L}_\varphi^2\alpha^2 + 4\|\boldsymbol{v}\|^2\alpha^2. \quad (26)$$

*Proof.* Let us first note that if $h$ is $\tau$-weakly convex, proper, and lower semicontinuous, then the proximity operator $\mathrm{prox}_{\alpha h}$ can be equivalently characterized via the optimality condition:

$$\boldsymbol{p} = \mathrm{prox}_{\alpha h}(\boldsymbol{x}) \quad \Longleftrightarrow \quad \boldsymbol{x} - \boldsymbol{p} \in \alpha\partial h(\boldsymbol{p}) \quad (27)$$

for all $\alpha \in (0, \tau^{-1})$. Consequently, since $\varphi$ is $\tau$-weakly convex, we have $\alpha^{-1}(F_v^\alpha(\boldsymbol{x}) - \alpha\boldsymbol{v}) \in \partial\varphi(\mathrm{prox}_{\alpha\varphi}(\boldsymbol{x} - \alpha\boldsymbol{v}))$. Using (22), this means

$$\varphi(\boldsymbol{x}) - \varphi(\mathrm{prox}_{\alpha\varphi}(\boldsymbol{x} - \alpha\boldsymbol{v})) \geq \frac{1}{\alpha}\langle F_v^\alpha(\boldsymbol{x}) - \alpha\boldsymbol{v}, F_v^\alpha(\boldsymbol{x})\rangle - \frac{\tau}{2}\|F_v^\alpha(\boldsymbol{x})\|^2.$$

In addition, due to $\boldsymbol{x}, \mathrm{prox}_{\alpha\varphi}(\boldsymbol{x} - \alpha\boldsymbol{v}) \in \mathrm{dom}\,\varphi$ and using Young's inequality, we have

$$\varphi(\boldsymbol{x}) - \varphi(\mathrm{prox}_{\alpha\varphi}(\boldsymbol{x} - \alpha\boldsymbol{v})) \leq \mathsf{L}_\varphi\|F_v^\alpha(\boldsymbol{x})\| \leq \mathsf{L}_\varphi^2\alpha + \frac{1}{4\alpha}\|F_v^\alpha(\boldsymbol{x})\|^2.$$

Using Young's inequality once more — $\langle \boldsymbol{v}, F_v^\alpha(\boldsymbol{x})\rangle \leq \alpha\|\boldsymbol{v}\|^2 + \frac{1}{4\alpha}\|F_v^\alpha(\boldsymbol{x})\|$ — it follows $\frac{1}{2}[\frac{1}{\alpha} - \tau]\|F_v^\alpha(\boldsymbol{x})\|^2 \leq \mathsf{L}_\varphi^2\alpha + \alpha\|\boldsymbol{v}\|^2$. The choice of $\alpha$ then readily implies (26). $\square$

Based on Lemma E.5, we now verify Lemma 3.5. Reusing the notation in Lemma E.5, we first observe

$$\boldsymbol{x}^{k+1} = \mathrm{prox}_{\alpha_k\varphi}(\boldsymbol{x}^k - \alpha_k\boldsymbol{g}^k) = \boldsymbol{x}^k - F_{\boldsymbol{g}^k}^{\alpha_k}(\boldsymbol{x}^k).$$

Hence, by (24), Lemma E.4, and (C.4), it follows

$$
\begin{aligned}
\mathbb{E}[\|\boldsymbol{x}^{k+1} - \boldsymbol{x}^k\|^2 \mid \mathcal{F}_k] &= \mathbb{E}[\|F_{\boldsymbol{g}^k}^{\alpha_k}(\boldsymbol{x}^k)\|^2 \mid \mathcal{F}_k] \leq 4\mathsf{L}_\varphi^2\alpha_k^2 + 4\alpha_k^2\mathbb{E}[\|\boldsymbol{g}^k\|^2 \mid \mathcal{F}_k] \\
&= 4\mathsf{L}_\varphi^2\alpha_k^2 + 4\alpha_k^2\|\nabla f(\boldsymbol{x}^k)\|^2 + 4\alpha_k^2\mathbb{E}[\|\nabla f(\boldsymbol{x}^k) - \boldsymbol{g}^k\|^2 \mid \mathcal{F}_k] \\
&\leq 4(2\mathsf{L} + \mathsf{C})\alpha_k^2 \cdot [f(\boldsymbol{x}^k) - \bar{f}] + 4(\mathsf{L}_\varphi^2 + \mathsf{D})\alpha_k^2 \\
&\leq 8(2\mathsf{L} + \mathsf{C})\alpha_k^2 \cdot [\mathrm{env}_{\theta\psi}(\boldsymbol{x}^k) - \bar{\psi}] + 4(((2\mathsf{L} + \mathsf{C})\theta + 1)\mathsf{L}_\varphi^2 + \mathsf{D})\alpha_k^2.
\end{aligned}
$$

This is exactly (12) in Lemma 3.5.

### E.6 Expected iteration complexity of prox-SGD

As a byproduct of our analysis, we provide the expected iteration complexity of prox-SGD below.

**Corollary E.6.** *Let $\{\boldsymbol{x}^k\}_{k \geq 0}$ be generated by* prox-SGD *and let the assumptions* (C.1)–(C.5) *be satisfied. Then, for $\theta \in (0, [3L + \tau]^{-1})$ and all $k$ with $\alpha_k \leq \min\{\frac{1}{2\tau}, \frac{1}{2(\theta^{-1} - [L + \tau])}\}$, it holds that*

$$\min_{k=0,\dots,T} \mathbb{E}[\|\nabla \mathrm{env}_{\theta\psi}(\boldsymbol{x}^k)\|^2] \leq \frac{\mathrm{env}_{\theta\psi}(\boldsymbol{x}^0) - \bar{\psi} + \mathsf{K} \sum_{k=0}^{T} \alpha_k^2}{\mathsf{L}\theta \sum_{k=0}^{T} \alpha_k}.$$

*Here, $\mathsf{K} > 0$ is defined in the proof. Consequently, if $\alpha_k = \frac{c}{(k+1)\log(k+2)}$ with some proper $c > 0$, then we have*

$$\min_{k=0,\dots,T} \mathbb{E}[\|\nabla \mathrm{env}_{\theta\psi}(\boldsymbol{x}^k)\|^2] \leq \mathcal{O}\left(\frac{\log(T+2)}{\sqrt{T+1}}\right).$$

*Proof.* Taking total expectation in Lemma 3.4 gives

$$\mathbb{E}[\mathrm{env}_{\theta\psi}(\boldsymbol{x}^{k+1}) - \bar{\psi}] \leq (1 + 4\mathsf{C}\theta^{-1}\alpha_k^2)\mathbb{E}[\mathrm{env}_{\theta\psi}(\boldsymbol{x}^k) - \bar{\psi}] \\ - \mathsf{L}\theta\alpha_k \mathbb{E}[\|\nabla \mathrm{env}_{\theta\psi}(\boldsymbol{x}^k)\|^2] + 2\alpha_k^2(\mathsf{CL}_\varphi^2 + \mathsf{D}\theta^{-1}). \tag{28}$$

Then, by unrolling this recursion and setting $c_1 := 4\mathsf{C}\theta^{-1}$ and $c_2 := 2(\mathsf{CL}_\varphi^2 + \mathsf{D}\theta^{-1})$, we have

$$\mathbb{E}[\mathrm{env}_{\theta\psi}(\boldsymbol{x}^{k+1}) - \bar{\psi}] \leq \left\{\prod_{j=0}^{k}(1 + c_1\alpha_j^2)\right\} \underbrace{[\mathrm{env}_{\theta\psi}(\boldsymbol{x}^0) - \bar{\psi}]}_{E_1} \\ + c_2 \sum_{j=0}^{k-1} \left\{\prod_{i=j+1}^{k}(1 + c_1\alpha_i^2)\right\}\alpha_j^2 + c_2\alpha_k^2.$$

Next, using $\log(1 + x) \leq x$ for all $x \geq 0$, we obtain the estimate

$$\prod_{i=j}^{k}(1 + c_1\alpha_i^2) = \exp\left(\sum_{i=j}^{k}\log(1 + c_1\alpha_i^2)\right) \leq \exp\left(\sum_{i=0}^{\infty} c_1\alpha_i^2\right) =: c_3$$

for all $j = 0, \dots, k$ and $k \in \mathbb{N}$. Thus, it follows $\mathbb{E}[\mathrm{env}_{\theta\psi}(\boldsymbol{x}^{k+1}) - \bar{\psi}] \leq c_3 E_1 + c_2 c_3 \sum_{j=0}^{k} \alpha_j^2$ and we can infer $\mathbb{E}[\mathrm{env}_{\theta\psi}(\boldsymbol{x}^k) - \bar{\psi}] \leq \mathsf{G}$ for all $k$ and some constant $\mathsf{G} > 0$. Invoking this bound in (28) yields

$$\mathbb{E}[\mathrm{env}_{\theta\psi}(\boldsymbol{x}^{k+1}) - \bar{\psi}] \leq \mathbb{E}[\mathrm{env}_{\theta\psi}(\boldsymbol{x}^k) - \bar{\psi}] - \mathsf{L}\theta\alpha_k \mathbb{E}[\|\nabla \mathrm{env}_{\theta\psi}(\boldsymbol{x}^k)\|^2] + \alpha_k^2 \underbrace{(c_1\mathsf{G} + c_2)}_{\mathsf{K}}.$$

Finally, summing the above recursion from $k = 0$ to $k = T$ and rearranging the terms, we have

$$\min_{k=0,\dots,T} \mathbb{E}[\|\nabla \mathrm{env}_{\theta\psi}(\boldsymbol{x}^k)\|^2] \leq \frac{\mathbb{E}[\mathrm{env}_{\theta\psi}(\boldsymbol{x}^0) - \bar{\psi}] + \mathsf{K} \sum_{k=0}^{T} \alpha_k^2}{\mathsf{L}\theta \sum_{k=0}^{T} \alpha_k}.$$

$\square$

## F   Results and proofs for Subsection 3.4

### F.1   Three stochastic model-based methods

Depending on the choice of the model function $f_{\boldsymbol{x}^k}(\cdot, \xi^k)$, we recover different stochastic optimization methods. Specifically, we can consider the following examples:

$$\begin{cases} f_{\boldsymbol{x}^k}(\boldsymbol{x}, \xi^k) = f(\boldsymbol{x}^k, \xi^k) + \langle \boldsymbol{s}(\boldsymbol{x}^k, \xi^k), \boldsymbol{x} - \boldsymbol{x}^k \rangle & \text{stochastic subgradient model,} \\ f_{\boldsymbol{x}^k}(\boldsymbol{x}, \xi^k) = f(\boldsymbol{x}, \xi^k) & \text{stochastic proximal point model,} \end{cases}$$

where $\boldsymbol{s}(\boldsymbol{x}^k, \xi^k) \in \partial f(\boldsymbol{x}^k, \xi^k)$ is a selected subgradient. In addition, if the weakly convex function $f$ has the composite form $f(\boldsymbol{x}, \xi) = h(c(\boldsymbol{x}, \xi), \xi)$, where $h(\cdot, \xi)$ is convex and Lipschitz continuous, $c(\cdot, \xi)$ is smooth and has Lipschitz continuous Jacobian, we can further cover the prox-linear model:

$$f_{\boldsymbol{x}^k}(\boldsymbol{x}, \xi^k) = h(c(\boldsymbol{x}^k, \xi^k) + \nabla c(\boldsymbol{x}^k, \xi^k)(\boldsymbol{x} - \boldsymbol{x}^k), \xi^k), \qquad \text{stochastic prox-linear model.}$$

## F.2 Proof of Lemma 3.9

Under our assumptions, the estimate [11, (4.8)] is valid. Furthermore, in [11, Lemma 4.1], it is shown that the assumptions (D.1) and (D.3) imply Lipschitz continuity of the mapping $f$ (with constant L). Hence, setting $\boldsymbol{x} = \boldsymbol{x}^k$ in [11, (4.8)], we have

$$
\begin{aligned}
\mathbb{E}[\|\boldsymbol{x}^{k+1} - \boldsymbol{x}^k\|^2 \mid \mathcal{F}_k] &\leq -\frac{2\alpha_k}{1 - \eta\alpha_k}\mathbb{E}[\psi(\boldsymbol{x}^{k+1}) - \psi(\boldsymbol{x}^k) \mid \mathcal{F}_k] + \frac{2L^2\alpha_k^2}{1 - \eta\alpha_k} \\
&\leq \frac{2(L + L_\varphi)\alpha_k}{1 - \eta\alpha_k}\mathbb{E}[\|\boldsymbol{x}^{k+1} - \boldsymbol{x}^k\| \mid \mathcal{F}_k] + \frac{2L^2\alpha_k^2}{1 - \eta\alpha_k} \\
&\leq \frac{1}{2}\mathbb{E}[\|\boldsymbol{x}^{k+1} - \boldsymbol{x}^k\|^2 \mid \mathcal{F}_k] + \frac{2(L + L_\varphi)^2\alpha_k^2}{(1 - \eta\alpha_k)^2} + \frac{2L^2\alpha_k^2}{1 - \eta\alpha_k},
\end{aligned}
$$

where we have used the Lipschitz continuity of $\psi$ in the second inequality, while the last inequality is due to Young's and Jensen inequalities. Rearranging terms gives

$$
\mathbb{E}[\|\boldsymbol{x}^{k+1} - \boldsymbol{x}^k\|^2 \mid \mathcal{F}_k] \leq \left(\frac{4(L + L_\varphi)^2}{(1 - \eta\alpha_k)^2} + \frac{4L^2}{1 - \eta\alpha_k}\right)\alpha_k^2.
$$

Thus, for all $k$ with $\alpha_k \leq 1/(2\eta)$, we obtain $\mathbb{E}[\|\boldsymbol{x}^{k+1} - \boldsymbol{x}^k\|^2 \mid \mathcal{F}_k] \leq (16(L + L_\varphi)^2 + 8L^2)\alpha_k^2$.

## F.3 Convergence of SMM: An extended setting

In the following, we show that the analysis and results presented in the previous section and in subsection 3.4 can be further strengthened and generalized. In particular, it is possible to work with assumptions that are more aligned with the conditions (C.1)–(C.4) for prox-SGD. We consider the assumptions:

(E.1) The stochastic model function $f_{\boldsymbol{x}}$ satisfies a one-sided accuracy property, i.e., we have $\mathbb{E}[f_{\boldsymbol{x}^k}(\boldsymbol{x}^k, \xi^k) \mid \mathcal{F}_k] = f(\boldsymbol{x}^k)$ for all $k$ and

$$
\mathbb{E}[f_{\boldsymbol{x}^k}(\boldsymbol{y}, \xi^k) - f(\boldsymbol{y}) \mid \mathcal{F}_k] \leq \frac{\tau}{2}\|\boldsymbol{x}^k - \boldsymbol{y}\|^2 \quad \forall\, \boldsymbol{y} \in \operatorname{dom}\varphi, \quad \forall\, k, \quad \text{almost surely.}
$$

(E.2) The function $\boldsymbol{y} \mapsto f_{\boldsymbol{x}^k}(\boldsymbol{y}, \xi^k)$ is $\eta$-weakly convex for all $k$.

(E.3) There exists a sequence $\{\boldsymbol{g}^k\}_{k\geq 0}$ and constants $C, D \geq 0$ such that $\boldsymbol{g}^k \in \partial f_{\boldsymbol{x}^k}(\boldsymbol{x}^k, \xi^k)$ and $\mathbb{E}[\|\boldsymbol{g}^k\|^2 \mid \mathcal{F}_k] \leq C[f(\boldsymbol{x}^k) - \bar{f}] + D$ almost surely for all $k$.

(E.4) The function $\varphi$ is $\rho$-weakly convex, proper, lower semicontinuous, and $L_\varphi$-Lipschitz continuous on $\operatorname{dom}\varphi$. In addition, $\varphi$ is bounded from below on $\operatorname{dom}\varphi$, i.e., we have $\varphi(\boldsymbol{x}) \geq \bar{\varphi}$ for all $\boldsymbol{x} \in \operatorname{dom}\varphi$.

(E.5) The step sizes $\{\alpha_k\}_{k\geq 0}$ satisfy $\sum_{k=0}^{\infty}\alpha_k = \infty$ and $\sum_{k=0}^{\infty}\alpha_k^2 < \infty$.

Concerning the regularity of $f$, we will work with *one* of the following scenarios:

(F.1) The mapping $f$ is bounded from below and $L_f$-Lipschitz continuous on $\operatorname{dom}\varphi$, i.e., there exists $\bar{f}$ such that $f(\boldsymbol{x}) \geq \bar{f}$ for all $\boldsymbol{x} \in \operatorname{dom}\varphi$ and we have $|f(\boldsymbol{x}) - f(\boldsymbol{y})| \leq L_f\|\boldsymbol{x} - \boldsymbol{y}\|$ for all $\boldsymbol{x}, \boldsymbol{y} \in \operatorname{dom}\varphi$.

(F.2) The function $f$ is bounded from below on $\mathbb{R}^n$, i.e., there is $\bar{f}$ such that $f(\boldsymbol{x}) \geq \bar{f}$ for all $\boldsymbol{x} \in \mathbb{R}^n$, and the gradient mapping $\nabla f$ is Lipschitz continuous (on $\mathbb{R}^n$) with modulus $L > 0$.

In contrast to subsection 3.4 and [11], we will not require explicit Lipschitz continuity of the model function $f_{\boldsymbol{x}}$. Instead — as we will verify now — convergence of SMM can be established by only assuming Lipschitz continuity or Lipschitz smoothness of the mapping $f$. As mentioned in Appendix F.2, the conditions (D.1) and (D.3) already imply Lipschitz continuity of $f$ (see again [11, Lemma 4.1]). Hence, assuming (F.1) can be more general. Moreover, the alternative assumption (F.2) allows to cover the case where $f$ is not Lipschitz continuous but sufficiently smooth. This situation appears more frequently in stochastic proximal point methods where the model function is chosen as $f_{\boldsymbol{x}^k}(\boldsymbol{x}, \xi^k) = f(\boldsymbol{x}, \xi^k)$. Assumption (E.2) is parallel to assumption (D.2). Assumption (E.3) is a mild variance-type condition that is similar to (C.4) and which can be weaker than (D.3). We now establish the core estimates provided in Lemma 3.8 and Lemma 3.9 under the more general assumptions (E.1)–(E.5) and (F.1) or (F.2). Our analysis is inspired by the proof of [11, Lemma 4.2] but is closer to [29, Appendix C].

*Lipschitz continuous $f$.* We first investigate core properties of SMM under (F.1). Let us define $\Psi_k(\boldsymbol{x}) := f_{\boldsymbol{x}^k}(\boldsymbol{x}, \xi^k) + \varphi(\boldsymbol{x})$ and $F_k(\boldsymbol{x}) := f_{\boldsymbol{x}^k}(\boldsymbol{x}, \xi^k) + \frac{\eta}{2}\|\boldsymbol{x} - \boldsymbol{x}^k\|^2$. Then, by assumption, the mapping $\boldsymbol{x} \mapsto \Psi_k(\boldsymbol{x}) + \frac{1}{2\alpha_k}\|\boldsymbol{x} - \boldsymbol{x}^k\|^2$ is strongly convex with parameter $\alpha_k^{-1} - \zeta$, where $\zeta = \eta + \rho$. Specifically, if $\alpha_k < \zeta^{-1}$ and by (14), we have

$$\Psi_k(\boldsymbol{x}) + \frac{1}{2\alpha_k}\|\boldsymbol{x} - \boldsymbol{x}^k\|^2 \geq \Psi_k(\boldsymbol{x}^{k+1}) + \frac{1}{2\alpha_k}\|\boldsymbol{x}^{k+1} - \boldsymbol{x}^k\|^2 + \frac{1}{2}[\alpha_k^{-1} - \zeta]\|\boldsymbol{x} - \boldsymbol{x}^{k+1}\|^2 \quad (29)$$

for all $\boldsymbol{x} \in \mathrm{dom}\,\varphi$ and due to the convexity of $F_k$ and the Lipschitz continuity of $f$, it holds that

$$\begin{aligned}
\Psi_k(\boldsymbol{x}) - \Psi_k(\boldsymbol{x}^{k+1}) &= [\psi(\boldsymbol{x}) - \psi(\boldsymbol{x}^{k+1})] + [f_{\boldsymbol{x}^k}(\boldsymbol{x}, \xi^k) - f(\boldsymbol{x})] + [f(\boldsymbol{x}^{k+1}) - f(\boldsymbol{x}^k)] \\
&\quad + [f(\boldsymbol{x}^k) - f_{\boldsymbol{x}^k}(\boldsymbol{x}^k, \xi^k)] + [F_k(\boldsymbol{x}^k) - F_k(\boldsymbol{x}^{k+1})] + \frac{\eta}{2}\|\boldsymbol{x}^{k+1} - \boldsymbol{x}^k\|^2 \\
&\leq [\psi(\boldsymbol{x}) - \psi(\boldsymbol{x}^{k+1})] + [f_{\boldsymbol{x}^k}(\boldsymbol{x}, \xi^k) - f(\boldsymbol{x})] + [f(\boldsymbol{x}^k) - f_{\boldsymbol{x}^k}(\boldsymbol{x}^k, \xi^k)] \\
&\quad + \langle \boldsymbol{g}^k, \boldsymbol{x}^k - \boldsymbol{x}^{k+1}\rangle + \mathsf{L}_f\|\boldsymbol{x}^{k+1} - \boldsymbol{x}^k\| + \frac{\eta}{2}\|\boldsymbol{x}^{k+1} - \boldsymbol{x}^k\|^2, \quad (30)
\end{aligned}$$

where we used $\boldsymbol{g}^k \in \partial f_{\boldsymbol{x}^k}(\boldsymbol{x}^k, \xi^k) = \partial F_k(\boldsymbol{x}^k)$. Upon taking conditional expectation and using the Cauchy-Schwarz and Young's inequality and (E.1), we obtain

$$\begin{aligned}
\mathbb{E}[\Psi_k(\boldsymbol{x}) - \Psi_k(\boldsymbol{x}^{k+1}) \mid \mathcal{F}_k] &\leq \mathbb{E}[\psi(\boldsymbol{x}) - \psi(\boldsymbol{x}^{k+1}) \mid \mathcal{F}_k] + 2\alpha_k \mathbb{E}[\|\boldsymbol{g}^k\|^2 \mid \mathcal{F}_k] + 2\mathsf{L}_f^2 \alpha_k \\
&\quad + \frac{\tau}{2}\|\boldsymbol{x}^k - \boldsymbol{x}\|^2 + \frac{1}{2}\left[\eta + \frac{1}{2\alpha_k}\right]\mathbb{E}[\|\boldsymbol{x}^{k+1} - \boldsymbol{x}^k\|^2 \mid \mathcal{F}_k].
\end{aligned}$$

Rearranging the terms in (29), this yields

$$\begin{aligned}
\mathbb{E}[\|\boldsymbol{x}^{k+1} - \boldsymbol{x}\|^2 \mid \mathcal{F}_k] \\
\leq \frac{1 + \alpha_k \tau}{1 - \alpha_k \zeta}\|\boldsymbol{x}^k - \boldsymbol{x}\|^2 + \frac{2\alpha_k}{1 - \alpha_k \zeta}\mathbb{E}[\psi(\boldsymbol{x}) - \psi(\boldsymbol{x}^{k+1}) \mid \mathcal{F}_k] \\
+ \frac{4\alpha_k^2}{1 - \alpha_k \zeta}\mathbb{E}[\|\boldsymbol{g}^k\|^2 \mid \mathcal{F}_k] + \frac{4\mathsf{L}_f^2 \alpha_k^2}{1 - \alpha_k \zeta} - \frac{1 - 2\alpha_k \eta}{2(1 - \alpha_k \zeta)}\mathbb{E}[\|\boldsymbol{x}^{k+1} - \boldsymbol{x}^k\|^2 \mid \mathcal{F}_k]. \quad (31)
\end{aligned}$$

Let us now define $\bar{\boldsymbol{x}}^k := \mathrm{prox}_{\theta\psi}(\boldsymbol{x}^k)$ for $\theta \in (0, \zeta^{-1})$. Then, due to $\boldsymbol{x}^k, \bar{\boldsymbol{x}}^k \in \mathrm{dom}\,\varphi$ and the Lipschitz continuity of $f$ and $\varphi$ and applying Young's inequality, it holds that

$$\begin{aligned}
f(\boldsymbol{x}^k) - \bar{f} \leq \psi(\boldsymbol{x}^k) - \bar{\psi} &\leq \mathrm{env}_{\theta\psi}(\boldsymbol{x}^k) + (\mathsf{L}_f + \mathsf{L}_\varphi)\|\boldsymbol{x}^k - \bar{\boldsymbol{x}}^k\| - \frac{1}{2\theta}\|\boldsymbol{x}^k - \bar{\boldsymbol{x}}^k\|^2 - \bar{\psi} \\
&\leq \mathrm{env}_{\theta\psi}(\boldsymbol{x}^k) - \bar{\psi} + 0.5\theta(\mathsf{L}_f + \mathsf{L}_\varphi)^2. \quad (32)
\end{aligned}$$

In addition, setting $\boldsymbol{x} = \bar{\boldsymbol{x}}^k$ in (31), using

$$\psi(\bar{\boldsymbol{x}}^k) = \mathrm{env}_{\theta\psi}(\boldsymbol{x}^k) - \frac{1}{2\theta}\|\boldsymbol{x}^k - \bar{\boldsymbol{x}}^k\|^2 \leq \psi(\boldsymbol{x}^{k+1}) + \frac{1}{2\theta}[\|\boldsymbol{x}^{k+1} - \boldsymbol{x}^k\|^2 - \|\boldsymbol{x}^k - \bar{\boldsymbol{x}}^k\|^2]$$

and combining (E.3) and (32), we obtain

$$\begin{aligned}
\mathbb{E}[\|\boldsymbol{x}^{k+1} - \bar{\boldsymbol{x}}^k\|^2 \mid \mathcal{F}_k] &\leq \left[1 - \frac{(\theta^{-1} - \zeta - \tau)\alpha_k}{1 - \alpha_k \zeta}\right]\|\boldsymbol{x}^k - \bar{\boldsymbol{x}}^k\|^2 + \frac{4\mathsf{C}\alpha_k^2}{1 - \alpha_k \zeta}[\mathrm{env}_{\theta\psi}(\boldsymbol{x}^k) - \bar{\psi}] \\
&\quad + \frac{\mathsf{G}\alpha_k^2}{1 - \alpha_k \zeta} - \frac{1 - 2[\eta + \theta^{-1}]\alpha_k}{2(1 - \alpha_k \zeta)}\mathbb{E}[\|\boldsymbol{x}^{k+1} - \boldsymbol{x}^k\|^2 \mid \mathcal{F}_k], \quad (33)
\end{aligned}$$

where $\mathsf{G} := 2\mathsf{C}\theta(\mathsf{L}_f + \mathsf{L}_\varphi)^2 + 4(\mathsf{L}_f^2 + \mathsf{D})$. Hence, using the definition of the Moreau envelope and (33), for $\theta \in (0, (\zeta + \tau)^{-1})$ and all $k$ with $\alpha_k \leq \min\{\frac{1}{2\zeta}, \frac{\theta}{2(1+\theta\eta)}\}$, it follows

$$\begin{aligned}
\mathbb{E}[\mathrm{env}_{\theta\psi}(\boldsymbol{x}^{k+1}) - \bar{\psi} \mid \mathcal{F}_k] \\
\leq [\mathrm{env}_{\theta\psi}(\boldsymbol{x}^k) - \bar{\psi}] + \frac{1}{2\theta}\left[\mathbb{E}[\|\boldsymbol{x}^{k+1} - \bar{\boldsymbol{x}}^k\|^2 \mid \mathcal{F}_k] - \|\bar{\boldsymbol{x}}^k - \boldsymbol{x}^k\|^2\right], \\
\leq \left[1 + \frac{4\mathsf{C}\alpha_k^2}{\theta}\right][\mathrm{env}_{\theta\psi}(\boldsymbol{x}^k) - \bar{\psi}] - \frac{1 - \theta(\zeta + \tau)}{2}\alpha_k\|\nabla\mathrm{env}_{\theta\psi}(\boldsymbol{x}^k)\|^2 + \frac{\mathsf{G}\alpha_k^2}{\theta}. \quad (34)
\end{aligned}$$

This gives an estimate similar to Lemma 3.8.

Next, let us consider an iteration $k$ with $\alpha_k \leq \min\{\frac{1}{2\zeta}, \frac{1}{2\eta}\}$. Using (31) with $\boldsymbol{x} = \boldsymbol{x}^k$ and Young's inequality, we obtain

$$\frac{1}{2}\mathbb{E}[\|\boldsymbol{x}^{k+1} - \boldsymbol{x}^k\|^2 \mid \mathcal{F}_k]$$

$$\leq 2\alpha_k\mathbb{E}[\psi(\boldsymbol{x}^k) - \psi(\boldsymbol{x}^{k+1}) \mid \mathcal{F}_k] + 4\alpha_k^2\mathbb{E}[\|\boldsymbol{g}^k\|^2 \mid \mathcal{F}_k] + 4\mathsf{L}_f^2\alpha_k^2$$

$$\leq 2(\mathsf{L}_f + \mathsf{L}_\varphi)\alpha_k\mathbb{E}[\|\boldsymbol{x}^{k+1} - \boldsymbol{x}^k\| \mid \mathcal{F}_k] + 4\mathsf{C}\alpha_k^2[\text{env}_{\theta\psi}(\boldsymbol{x}^k) - \bar\psi] + \mathsf{G}\alpha_k^2$$

$$\leq \frac{1}{4}\mathbb{E}[\|\boldsymbol{x}^{k+1} - \boldsymbol{x}^k\|^2 \mid \mathcal{F}_k] + 4\mathsf{C}\alpha_k^2[\text{env}_{\theta\psi}(\boldsymbol{x}^k) - \bar\psi] + [\mathsf{G} + 4(\mathsf{L}_f + \mathsf{L}_\varphi)^2]\alpha_k^2.$$

Rearranging the terms yields $\mathbb{E}[\|\boldsymbol{x}^{k+1} - \boldsymbol{x}^k\|^2 \mid \mathcal{F}_k] \leq 16\mathsf{C}\alpha_k^2[\text{env}_{\theta\psi}(\boldsymbol{x}^k) - \bar\psi] + 4[\mathsf{G} + 4(\mathsf{L}_f + \mathsf{L}_\varphi)^2]\alpha_k^2$, which is similar to the estimates in Lemma 3.9 (indeed, it is more similar to Lemma 3.5).

With these estimates and by following exactly the derivations in Subsection 3.3, we can show $\mathbb{E}[\|\nabla\text{env}_{\theta\psi}(\boldsymbol{x}^k)\|] \to 0$ and $\|\nabla\text{env}_{\theta\psi}(\boldsymbol{x}^k)\| \to 0$ almost surely.

*Lipschitz smooth $f$.* We continue our discussion under (F.2). Using the Lipschitz continuity of $\nabla f$, the estimate (30) changes to

$$\Psi_k(\boldsymbol{x}) - \Psi_k(\boldsymbol{x}^{k+1}) \leq [\psi(\boldsymbol{x}) - \psi(\boldsymbol{x}^{k+1})] + [f_{\boldsymbol{x}^k}(\boldsymbol{x}, \xi^k) - f(\boldsymbol{x})] + [f(\boldsymbol{x}^k) - f_{\boldsymbol{x}^k}(\boldsymbol{x}^k, \xi^k)]$$

$$+ \langle \boldsymbol{g}^k - \nabla f(\boldsymbol{x}^k), \boldsymbol{x}^k - \boldsymbol{x}^{k+1}\rangle + \frac{\mathsf{L} + \eta}{2}\|\boldsymbol{x}^{k+1} - \boldsymbol{x}^k\|^2.$$

Taking conditional expectation and applying Young's inequality and (E.1), this yields

$$\mathbb{E}[\Psi_k(\boldsymbol{x}) - \Psi_k(\boldsymbol{x}^{k+1}) \mid \mathcal{F}_k] \leq \mathbb{E}[\psi(\boldsymbol{x}) - \psi(\boldsymbol{x}^{k+1}) \mid \mathcal{F}_k] + 2\alpha_k\mathbb{E}[\|\boldsymbol{g}^k\|^2 \mid \mathcal{F}_k] + 2\alpha_k\|\nabla f(\boldsymbol{x}^k)\|^2$$

$$+ \frac{\tau}{2}\|\boldsymbol{x}^k - \boldsymbol{x}\|^2 + \frac{1}{2}\left[\mathsf{L} + \eta + \frac{1}{2\alpha_k}\right]\mathbb{E}[\|\boldsymbol{x}^{k+1} - \boldsymbol{x}^k\|^2 \mid \mathcal{F}_k].$$

By Lemma E.4 and (24), we can further infer $f(\boldsymbol{x}) - \bar f \leq 2[\text{env}_{\theta\psi}(\boldsymbol{x}) - \bar\psi] + \mathsf{L}_\varphi^2\theta$ and $\|\nabla f(\boldsymbol{x})\|^2 \leq 2\mathsf{L}[f(\boldsymbol{x}) - \bar f]$ for all $\boldsymbol{x} \in \text{dom}\,\varphi$. (Notice that the conditions (C.1)–(C.3) and (E.4) and (F.2) are identical). Hence, similar to (31), it follows

$$\mathbb{E}[\|\boldsymbol{x}^{k+1} - \boldsymbol{x}\|^2 \mid \mathcal{F}_k]$$

$$\leq \frac{1 + \alpha_k\tau}{1 - \alpha_k\zeta}\|\boldsymbol{x}^k - \boldsymbol{x}\|^2 + \frac{2\alpha_k}{1 - \alpha_k\zeta}\mathbb{E}[\psi(\boldsymbol{x}) - \psi(\boldsymbol{x}^{k+1}) \mid \mathcal{F}_k] + \frac{\tilde{\mathsf{G}}\alpha_k^2}{1 - \alpha_k\zeta}$$

$$+ \frac{8(\mathsf{C} + 2\mathsf{L})\alpha_k^2}{1 - \alpha_k\zeta}[\text{env}_{\theta\psi}(\boldsymbol{x}^k) - \bar\psi] - \frac{1 - 2\alpha_k(\mathsf{L} + \eta)}{2(1 - \alpha_k\zeta)}\mathbb{E}[\|\boldsymbol{x}^{k+1} - \boldsymbol{x}^k\|^2 \mid \mathcal{F}_k], \quad (35)$$

where $\tilde{\mathsf{G}} = 4((\mathsf{C} + 2\mathsf{L})\mathsf{L}_\varphi^2 + \mathsf{D})$ and $\zeta = \eta + \rho$. At this point, we can fully mimic our earlier calculations. In particular, similar to (34), for $\theta \in (0, (\zeta + \tau)^{-1})$ and all $k$ with $\alpha_k \leq \min\{\frac{1}{2\zeta}, \frac{\theta}{2(1+\theta(\mathsf{L}+\eta))}\}$, we obtain

$$\mathbb{E}[\text{env}_{\theta\psi}(\boldsymbol{x}^{k+1}) - \bar\psi \mid \mathcal{F}_k] \leq \left[1 + \frac{8(\mathsf{C} + 2\mathsf{L})\alpha_k^2}{\theta}\right][\text{env}_{\theta\psi}(\boldsymbol{x}^k) - \bar\psi]$$

$$- \frac{1 - \theta(\zeta + \tau)}{2}\alpha_k\|\nabla\text{env}_{\theta\psi}(\boldsymbol{x}^k)\|^2 + \frac{\tilde{\mathsf{G}}\alpha_k^2}{\theta}.$$

Setting $\boldsymbol{x} = \boldsymbol{x}^k$ in (35) and using the Lipschitz continuity of $\varphi$ and $\nabla f$ and Young's inequality, it holds that

$$\frac{1}{2}\mathbb{E}[\|\boldsymbol{x}^{k+1} - \boldsymbol{x}^k\|^2 \mid \mathcal{F}_k] \leq 2\alpha_k\mathbb{E}[\psi(\boldsymbol{x}^k) - \psi(\boldsymbol{x}^{k+1}) \mid \mathcal{F}_k]$$

$$+ 8(\mathsf{C} + 2\mathsf{L})\alpha_k^2[\text{env}_{\theta\psi}(\boldsymbol{x}^k) - \bar\psi] + \tilde{\mathsf{G}}\alpha_k^2$$

$$\leq 2\mathsf{L}_\varphi\alpha_k\mathbb{E}[\|\boldsymbol{x}^{k+1} - \boldsymbol{x}^k\| \mid \mathcal{F}_k] + \mathsf{L}\alpha_k\mathbb{E}[\|\boldsymbol{x}^{k+1} - \boldsymbol{x}^k\|^2 \mid \mathcal{F}_k]$$

$$- 2\alpha_k\mathbb{E}[\langle\nabla f(\boldsymbol{x}^k), \boldsymbol{x}^{k+1} - \boldsymbol{x}^k\rangle \mid \mathcal{F}_k] + 8(\mathsf{C} + 2\mathsf{L})\alpha_k^2[\text{env}_{\theta\psi}(\boldsymbol{x}^k) - \bar\psi] + \tilde{\mathsf{G}}\alpha_k^2$$

$$\leq \frac{\mathbb{E}[\|\boldsymbol{x}^{k+1} - \boldsymbol{x}^k\|^2 \mid \mathcal{F}_k]}{4} + 16\alpha_k^2\|\nabla f(\boldsymbol{x}^k)\|^2 + 8(\mathsf{C} + 2\mathsf{L})\alpha_k^2[\text{env}_{\theta\psi}(\boldsymbol{x}^k) - \bar\psi] + [\tilde{\mathsf{G}} + 16\mathsf{L}_\varphi^2]\alpha_k^2$$

$$\leq \frac{1}{4}\mathbb{E}[\|\boldsymbol{x}^{k+1} - \boldsymbol{x}^k\|^2 \mid \mathcal{F}_k] + 8(\mathsf{C} + 10\mathsf{L})\alpha_k^2[\mathrm{env}_{\theta\psi}(\boldsymbol{x}^k) - \bar{\psi}] + [\tilde{\mathsf{G}} + 16(1 + 2\mathsf{L}\theta)\mathsf{L}_\varphi^2]\alpha_k^2$$

for all $k$ with $\alpha_k \leq \min\{\frac{1}{2\zeta}, \frac{1}{8\mathsf{L}}, \frac{1}{2(\mathsf{L}+\eta)}\}$. As before, this establishes variants of Lemma 3.8 and Lemma 3.9 (more precisely, Lemma 3.5) and allows to follow the derivations in Subsection 3.3 to establish convergence results.

Finally, we summarize all the above observations in the following corollary.

**Corollary F.1.** *Let us consider the family of stochastic model-based methods* (14) *for the problem* (13) *under assumptions* (E.1)–(E.5) *and* (F.1) *or* (F.2)*. Then, for all* $\theta \in (0, (\eta + \rho + \tau)^{-1})$*, we have* $\lim_{k\to\infty}\mathbb{E}[\|\nabla\mathrm{env}_{\theta\psi}(\boldsymbol{x}^k)\|] = 0$ *and* $\lim_{k\to\infty}\|\nabla\mathrm{env}_{\theta\psi}(\boldsymbol{x}^k)\| = 0$ *almost surely.*

# G  Comparison: related literature

Table 1: Summary and comparison of related and representative literature.

| SGD | Assumptions | Convergence | |
|---|---|---|---|
| | | in expectation | almost surely |
| [3] | (A.1), (A.2), (A.4), bounded variance ((A.3) with $\mathsf{C} = 0$) | ✗ | ✓ |
| [28] | (A.1), (A.4) $f$ is coercive ($\implies$ (A.2)) $f$ is Lipschitz $\liminf_{\|x\|\to\infty}\|\nabla f(x)\| > 0$ bounded variance ((A.3) with $\mathsf{C} = 0$) | ✗ | ✓ |
| [6] | (A.1), (A.2), (A.4) $f$ is twice differentiable $x \mapsto \nabla^2 f(x)\nabla f(x)$ is Lipschitz bounded variance ((A.3) with $\mathsf{C} = 0$) | ✓ | ✗ |
| This work | (A.1)–(A.4) | ✓ | ✓ |

| RR | Assumptions | Convergence | |
|---|---|---|---|
| | | in expectation | almost surely |
| [25] | (B.1)–(B.2) | ✗ | ✓ |
| This work | (B.1)–(B.2) | ✓ | ✓ |

| prox-SGD | Assumptions | Convergence | |
|---|---|---|---|
| | | in expectation | almost surely |
| [27] | (C.1), (C.3), (C.5) $\{\boldsymbol{x}^k\}_{k\geq 0}$ is surely bounded almost surely bounded noise | ✗ | ✓ |
| This work | (C.1)–(C.5) | ✓ | ✓ |

| SMM | Assumptions | Convergence | |
|---|---|---|---|
| | | in expectation | almost surely |
| [14] | (D.3), (D.5) surely one-sided accuracy ($\implies$ (D.1)) $\varphi$ is convex ($\implies$ (D.2))) $\{\boldsymbol{x}^k\}_{k\geq 0}$ is surely bounded (compact constraint) density / Sard-type condition | ✗ | ✓ |
| This work | (D.1)–(D.5) | ✓ | ✓ |

# H  Non-asymptotic complexity vs. asymptotic convergence

In this subsection, we provide several additional arguments and illustrations that can help to explain and illuminate the potential differences between typical finite-step complexity rates and asymptotic convergence results — as obtained in Theorem 2.1. To this end, we consider the standard optimization problem

$$\min_{\boldsymbol{x} \in \mathbb{R}^n} f(\boldsymbol{x}), \tag{36}$$

where $f : \mathbb{R}^n \to \mathbb{R}$ is a given, smooth, and nonconvex function. As motivated in the introduction, complexity bounds for the nonconvex problem (36) typically take the form

$$\min_{k=0,\dots,T} \mathbb{E}[\|\nabla f(\boldsymbol{x}^k)\|^2] = \mathcal{O}((T+1)^{-\frac{1}{2}}) \quad \text{or} \quad \mathbb{E}[\|\nabla f(\boldsymbol{x}^{\bar{k}})\|^2] \le \mathcal{O}((T+1)^{-\frac{1}{2}}). \tag{37}$$

Here, $T$ denotes the total number of iterations, the index $\bar{k}$ is sampled uniformly at random from $\{0, \dots, T\}$, and the iterates $\{\boldsymbol{x}^k\}_{k \ge 0}$ are assumed to be generated by the stochastic gradient descent method, see [17, 6, 24, 23]. Similar (deterministic) complexity results are also available for the basic gradient descent method, [2], and many other related algorithmic schemes, [18, 11, 30, 32]. In particular, for the gradient descent method, the complexity bounds (37) can be strengthened to

$$\min_{k=0,\dots,T} \|\nabla f(\boldsymbol{x}^k)\|^2 = \mathcal{O}((T+1)^{-1}), \tag{38}$$

see, e.g., [2, Theorem 10.15].

While the complexity bounds shown in (37) (and (38)) allow to capture and characterize the overall trend of the minimization procedure during the first $T$ iterations, they cannot fully justify a common practice in stochastic optimization: the last iterate $\boldsymbol{x}^T$ is returned as final output of the algorithm. In fact, even as $T$ increases, the term $\mathbb{E}[\|\nabla f(\boldsymbol{x}^T)\|]$ can be arbitrarily large whereas the complexity measure $\min_{k=0,\dots,T} \mathbb{E}[\|\nabla f(\boldsymbol{x}^k)\|^2]$ decreases at its respective rate. The asymptotic convergence results

$$\lim_{k \to \infty} \mathbb{E}[\|\nabla f(\boldsymbol{x}^k)\|] = 0 \quad \text{or} \quad \lim_{k \to \infty} \|\nabla f(\boldsymbol{x}^k)\| = 0 \quad \text{almost surely,}$$

can provide additional information: as $\{\mathbb{E}[\|\nabla f(\boldsymbol{x}^k)\|]\}_{k \ge 0}$ converges to zero, the term $\mathbb{E}[\|\nabla f(\boldsymbol{x}^T)\|]$ will stay small (below any predefined threshold) for all $T$ sufficiently large. In tandem with the complexity bounds (37), this supports common output strategies that return the last iterate $\boldsymbol{x}^T$ — at least for large $T$. Hence, both non-asymptotic and asymptotic convergence analyses are important and informative — especially in the nonconvex and stochastic setting — and the combination of non-asymptotic and asymptotic convergence guarantees can paint a more complete picture of the convergence behavior of stochastic optimization methods.

We continue with a specific example that can illustrate the mentioned discrepancies and differences between non-asymptotic complexity and asymptotic convergence results. In particular, we construct a nonconvex function $f : \mathbb{R} \to \mathbb{R}$, a corresponding step size sequence $\{\alpha_k\}_{k \ge 0}$, and an initial point $\boldsymbol{x}^0 \in \mathbb{R}$, for which the standard gradient descent method generates a sequence of iterates $\{\boldsymbol{x}^k\}_{k \ge 0}$ that satisfies the non-asymptotic complexity bound (38), but we can *not* observe asymptotic convergence $f'(\boldsymbol{x}^k) \to 0$.

Let $0 \le \nu < \kappa$ be given parameters. We consider the functions

$$c(x) := \begin{cases} e^{-\frac{1}{x}} & \text{if } x > 0, \\ 0 & \text{otherwise,} \end{cases} \quad \text{and} \quad \bar{c}_{\kappa,\nu}(x) := \frac{c(\kappa^2 - x)}{c(\kappa^2 - x) + c(x - \nu^2)}.$$

The mappings $c$ and $\bar{c}_{\kappa,\nu}$ are obviously $C^\infty$ and we have $\bar{c}_{\kappa,\nu}(x) = 0$ for all $x \ge \kappa^2$ and $\bar{c}_{\kappa,\nu}(x) = 1$ for all $x \le \nu^2$. Moreover, it holds that

$$c'(x) = \begin{cases} \frac{1}{x^2} e^{-\frac{1}{x}} & \text{if } x > 0, \\ 0 & \text{otherwise,} \end{cases} \quad \text{and} \quad \bar{c}'_{\kappa,\nu}(x) = -\frac{c'(\kappa^2 - x)c(x - \nu^2) + c(\kappa^2 - x)c'(x - \nu^2)}{(c(\kappa^2 - x) + c(x - \nu^2))^2}.$$

We now define $\kappa_j := \frac{1}{4j(2j+1)}$, $\nu_j := \frac{1}{8j(2j+1)}$, $g(x) := x - \frac{1}{2}x^2$, $\gamma_j(x) := g(x)\bar{c}_{\kappa_j,\nu_j}((x - \frac{1}{2j})^2)$, and

$$f(x) := \begin{cases} h(x) + \sum_{k=1}^\infty \gamma_k(x) & \text{if } x \ne 0, \\ 0 & \text{if } x = 0, \end{cases} \quad \text{and} \quad h(x) := \begin{cases} \frac{1}{2}x^2 & \text{if } x \ge 0, \\ 8x^2(8x^2 - 1) & \text{if } x < 0. \end{cases}$$

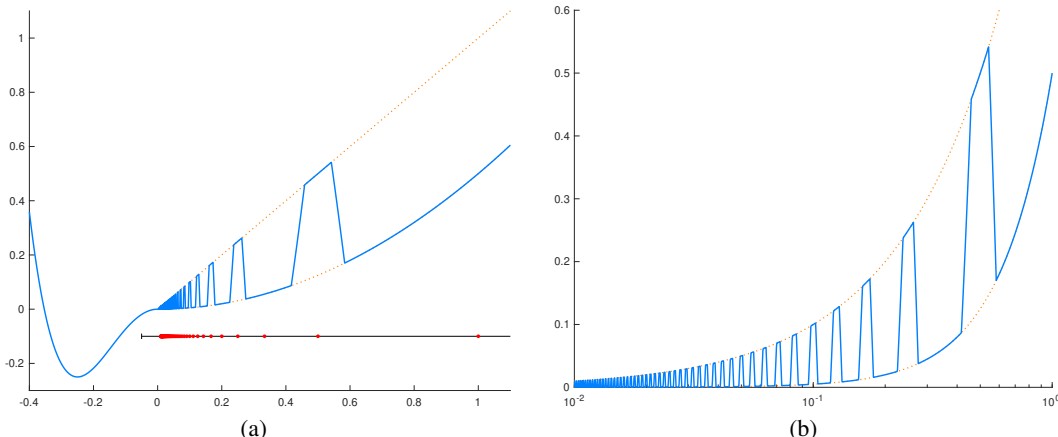

(a)                                       (b)

Figure 1: Plot of $f$. The function $f$ is depicted using a blue color. The dotted orange lines correspond to the functions $x \mapsto x$ and $x \mapsto \frac{1}{2}x^2$, respectively. The read points in subfigure (a) show the iterates $\boldsymbol{x}^k$, $k \in \mathbb{N}$. Subfigure (b) shows a logarithmic plot of $f$ on $[0.01, 1]$.

An exemplary plot of the function $f$ is shown in Figure 1. The function $f$ is continuous on $\mathbb{R}$ and smooth for all $x \neq 0$. We now want to run gradient descent on $f$ with initial point $\boldsymbol{x}^0 = 1$ and diminishing step sizes

$$\alpha_k := \frac{1}{k+2}, \quad \text{if } k \text{ is even} \quad \text{and} \quad \alpha_k := \frac{1}{(k+1)(k+2)}, \quad \text{if } k \text{ is odd.}$$

Then, it follows

$$\boldsymbol{x}^k = \frac{1}{k+1} \quad \text{and} \quad f'(\boldsymbol{x}^k) = \begin{cases} 1 & \text{if } k \text{ is even,} \\ \frac{1}{k+1} & \text{if } k \text{ is odd.} \end{cases} \tag{39}$$

We now verify this statement by induction. We first notice that the functions $\{\gamma_k\}_{k\geq 0}$ and derivatives $\{\gamma_k'\}_{k\geq 0}$ have disjoint supports $\left[\frac{1}{2k} - \frac{1}{4k(2k+1)}, \frac{1}{2k} + \frac{1}{4k(2k+1)}\right] = \left[\frac{1}{2}\left(\frac{1}{2k} + \frac{1}{2k+1}\right), \frac{1}{2}\left(\frac{3}{2k} - \frac{1}{2k+1}\right)\right]$ with center point $\frac{1}{2k}$, $k \in \mathbb{N}$. More specifically, we have

$$\gamma_k(x) = \begin{cases} 0 & \text{if } |x - \frac{1}{2k}| \geq \frac{1}{4k(2k+1)}, \\ g(x) & \text{if } |x - \frac{1}{2k}| \leq \frac{1}{8k(2k+1)}, \end{cases} \quad \text{and} \quad \gamma_k'(x) = \begin{cases} 0 & \text{if } |x - \frac{1}{2k}| \geq \frac{1}{4k(2k+1)}, \\ g'(x) & \text{if } |x - \frac{1}{2k}| \leq \frac{1}{8k(2k+1)}. \end{cases}$$

Hence, (39) clearly holds for the base case $k = 0$. Let us now assume that the induction hypothesis (39) is true for some $k$ and let $k + 1 = 2j$, $j \in \mathbb{N}$, be an even number. Then, we obtain

$$\begin{aligned}
\boldsymbol{x}^{k+1} &= \boldsymbol{x}^k - \alpha_k f'(\boldsymbol{x}^k) \\
&= \frac{1}{k+1} - \frac{1}{(k+1)(k+2)} \cdot \left[\frac{1}{k+1} + \gamma_j'\left(\frac{1}{k+1}\right)\right] \\
&= \frac{1}{k+1} - \frac{1}{(k+1)(k+2)} \cdot 1 = \frac{1}{k+2}.
\end{aligned}$$

Similarly, if $k + 1 = 2j + 1$, $j \in \mathbb{N}$, is odd, we then have $\frac{1}{2j} - \frac{1}{k+1} = \frac{1}{2j(2j+1)} > \frac{1}{4j(2j+1)}$ and $\frac{1}{k+1} - \frac{1}{2j+2} = \frac{1}{(2j+1)(2j+2)} > \frac{1}{(2j+2)(2j+3)}$. This yields

$$\boldsymbol{x}^{k+1} = \boldsymbol{x}^k - \alpha_k f'(\boldsymbol{x}^k) = \frac{1}{k+1} - \frac{1}{k+2} \cdot \frac{1}{k+1} = \frac{1}{k+2}.$$

This finishes the proof of (39). We further notice

$$\sum_{k=0}^{\infty} \alpha_k \geq \sum_{k=0}^{\infty} \alpha_{2k} = \sum_{k=0}^{\infty} \frac{1}{2(k+1)} = \infty \quad \text{and} \quad \sum_{k=0}^{\infty} \alpha_k^2 < \infty.$$

Consequently, the step sizes $\{\alpha_k\}_k$ satisfy all standard requirements. In addition, we have

$$\sum_{k=0}^{\infty} \alpha_k |f'(\boldsymbol{x}^k)|^2 \leq \sum_{k=0}^{\infty} \frac{1}{(k+1)(k+2)} < \infty \quad \text{and} \quad \min_{k=0,\ldots,T} |f'(\boldsymbol{x}^k)|^2 \leq \frac{1}{T^2}.$$

Thus, the complexity results in (38) obviously hold, but the gradient values $f'(\boldsymbol{x}^k)$ do not converge to zero. In particular, for even $T$, the last iterate $\boldsymbol{x}^T$ satisfies $f'(\boldsymbol{x}^T) = 1$ which obstructs interpretability of $\boldsymbol{x}^T$ and of the complexity bounds $\min_{k=0,\ldots,T} |f'(\boldsymbol{x}^k)|^2 \leq \varepsilon$ or $|f'(\boldsymbol{x}^{\bar{k}})|^2 \leq \varepsilon$. Notice that the mapping $f$ is not Lipschitz smooth around $x = 0$ and hence, the convergence results in Theorem 2.1 are not applicable here. Of course, these observations have even higher significance in the stochastic setting, when evaluation of the bounds (37) is generally restrictive or not possible within the algorithmic procedure.

We conclude and summarize our discussion with a comment by Francesco Orabona on non-asymptotic and asymptotic convergence analyses for SGD ([33], blog post: "*Almost sure convergence of SGD on smooth non-convex functions*", section 5 and 6, Oct. 05, 2020):

*"Note that the 20-30 years ago there were many papers studying the asymptotic convergence of SGD and its variants in various settings. Then, the taste of the community changed moving from asymptotic convergence to finite-time rates. As it often happens when a new trend takes over the previous one, new generations tend to be oblivious to the old results and proof techniques. The common motivation to ignore these past results is that the finite-time analysis is superior to the asymptotic one, but this is clearly false (ask a statistician!). It should be instead clear to anyone that both analyses have pros and cons."*