# OpenReview forum: "A Unified Convergence Theorem for Stochastic Optimization Methods"
_NeurIPS.cc/2022/Conference — NeurIPS 2022 Accept_

### Official Review · Reviewer_T2QD · 2022-06-23

**Rating:** 6
**Confidence:** 3
**Soundness:** 3 good
**Presentation:** 4 excellent
**Contribution:** 3 good

**Summary:**

The authors analyze the convergence of SGD-like methods such that the norm of the gradient in the last iterate convergence to zero in expectation and almost surely. They provide the general framework to analyze the convergence for SGD-like methods, including SGD, Random Reshuffling, Prox-SGD, and Model-Based methods. Using this framework, they show the convergence of these methods.

**Questions:**

Suggestion: In the proof of Theorem 2.1, I think that it is possible not to differentiate $q = 1$ and $q > 1$. Can we in (16) use the boundness of $\beta_t$ and $\mu_k,$ and then use the same arguments as for the case $q = 1$? This can simplify the proof.

**Strengths And Weaknesses:**

i) The authors prove the main Theorem 2.1, which generalizes results from, e.g. https://parameterfree.com/2020/10/05/almost-sure-convergence-of-sgd-on-smooth-non-convex-functions/
or Gradient convergence in gradient methods with errors. Bertsekas et al.

Strengths: The generalization can be useful and give insights into the community.

Weaknesses: Yes, it can help to analyze a broad family of methods, but I think that the paper doesn't have enough examples to be sure that Theorem 2.1 is the "Unified Convergence Theorem."

ii) As the authors noted, the last-iterate convergence of SGD and RR was analyzed before, so, qualitatively, the paper's contribution is the analysis of Prox-SGD and SMM.

Strengths: Even the fact that it is possible to prove the convergence of Prox-SGD is interesting.

Weaknesses: To prove the convergence, the authors use Assumption (C.3) that $\varphi$ is $L$-Lipschitz. For instance, one of the most popular regularizers $\varphi(x) = \frac{1}{2} ||x||^2_2$ is not $L$-Lipschitz.

I think the paper is good. It provides proof that Prox-SGD and SMM converge. At the same time, I'm not sure that Theorem 2.1 can be considered the "Unified Convergence Theorem," but it can be helpful for the community.

---

> ### Author Response · Authors · 2022-08-02
> **Authors' Response**
>
> Thank you very much for the supportive and valuable comments. We address the reviewer's concerns and questions in a point-by-point manner below. Major changes in the revised manuscript are highlighted in blue.
>
> * **(i)** **About ''Unified Convergence Theorem'':** The reviewer raised a concern on whether Theorem 2.1 can be considered as a ''*unified convergence theorem*''. We believe that this question concerns whether we have potentially overclaimed the generality of our convergence theorem. Here, we would like to briefly explain our understanding of the term ''unified''. By ''unified", we mean that we can derive the convergence results of SGD, RR, prox-SGD, SMM, and possibly other stochastic optimization methods ''in a unified way'', i.e., through checking the conditions listed in Theorem 2.1. This is precisely what we have demonstrated in our manuscript for the mentioned stochastic methods. Indeed, the terminology ''unified'' is often used in the optimization community to refer to an analysis/approach that can unify several existing ones. Hence, the term ''unified convergence theorem'' should be distinguished from a ''universal convergence theorem''. However, if the reviewer is still concerned about this notation, we are more than happy to change it, e.g., to ''general convergence theorem'' or to other suitable terminologies.
>
>
>     Moreover, to further illustrate the broad applicability and flexibility of Theorem 2.1, we have derived additional convergence results for SMM and the stochastic proximal point method for solving (13) when $f$ is sufficiently smooth. We also extended the results in Corollary 3.10 by replacing (D.3) with a more general Lipschitz continuity assumption of $f$ and a more common ABC-type variance condition; see Remark 3.11 and Appendix F.3 in the revised manuscript. These results are in line with the convergence results obtained for prox-SGD and, to the best of our knowledge, appear to be new.
>
> * **(ii)** **About the regularizer $\frac{1}{2}||x||^2$:** One often absorbs smooth regularizers of the form ''$x \mapsto \frac{1}{2}||x||^2$'' in $f$ rather than using a proximal operator to tackle it directly. Therefore, this does not immediately contradict our Lipschitz assumption on $\varphi$.
>
> * **(iii)** **Suggestions on simplifying the proof:** Thank you for this nice observation! We have simplified the proof by considering $q\geq 1$ directly, which makes it more accessible to the reader. In addition, we realized that statement (i) of Theorem 2.1 can be further generalized and that the previous requirement ''$\mu_k \to 0$'' can be weakened to ''{$\mu_k$}$_{k\geq 0}$ is bounded''. This does not require any significant adjustments of the proof of Theorem 2.1. All changes are shown in blue color in Appendix A in the revised manuscript.
>
> We hope that our response and revisions are satisfactory to the reviewer and that all concerns have been addressed appropriately.

---

### Official Review · Reviewer_qySk · 2022-07-07

**Rating:** 7
**Confidence:** 3
**Soundness:** 3 good
**Presentation:** 3 good
**Contribution:** 2 fair

**Summary:**

This work provides a unified theorem for analyzing several stochastic optimization methods. Both expected and almost sure convergence are derived. For applications, the authors recover the convergence results for stochastic gradient descent (SGD) and random reshuffling (RR). In addition, this paper also obtains the convergence result for the stochastic proximal gradient method (prox-SGD) by using the proposed framework.

**Questions:**

Is there a typo in assumption (B.2)? Why do you require the sum of alpha^3 < infinity. The description for random reshuffling is rather rough, I would suggest the authors provide more details in the appendix for this method.

Some minors: in line 54 the hessian and gradient, in line 350 analyses.


EDIT: after the authors' response, I increased my score to 7.

**Limitations:**

Yes.

**Strengths And Weaknesses:**

This paper is well written and easy to follow in general. The structure of the paper is good and provides some new insights into the convergence results for the stochastic optimization methods. The core theorem 2.1 is very important and its proof makes sense to me. Furthermore, the prox-SGD is analyzed via this framework to obtain some convergence results.

The main weakness is that the proposed method only provides an asymptotic convergence result, which is less informative compared to the finite-time type of bounds. In addition, the function value convergence might be also interested.

---

> ### Author Response · Authors · 2022-08-02
> **Authors' Response: Part II**
>
> **B.** **Function value convergence:** The function value convergence can often be derived from the corresponding gradient norm convergence result. Again, let us take SGD as an example to convey the main idea. Applying the supermartingale convergence theorem to its recursion (6) yields that the sequence of function values converges to a finite random variable. Since the gradient norms {$||\nabla f(x^k)||$}$_{k \geq 0}$ converge to zero almost surely, every accumulation point will be a stationary point almost surely. Then, one immediately has $f(x^k)\to \bar f$ almost surely. Here, $\bar f$ is a finite random variable which generates realizations that are the function values of stationary points. Similar convergence for the expected function values can be derived by further applying the dominated convergence theorem.
>
> **C.** **About $\sum_{k=0}^\infty\alpha_k^3<\infty$ and RR:** This is not a typo. Since RR has smaller variance than SGD, it can tolerate slower diminishing step sizes. Furthermore, we have added more detailed descriptions and motivations for RR in Appendix D in the revised manuscript.
>
> **D.** **Minor comments:** Please notice that the word ''Hessian'' is typically capitalized since this terminology comes from and is connected to the German mathematician Ludwig Otto Hesse (this is similar to ''Newton'' in ''Newton's method''). The word ''analyses'' is the plural form of ''analysis'' and should be correct here.
>
> We hope that our response and revisions are satisfactory to the reviewer and that all concerns have been addressed appropriately.

---

> > ### Comment · Reviewer_qySk · 2022-08-08
> > **Thanks for addressing my questions**
> >
> > Thank you for the detailed response. Sorry I could not reply timely. I do not have any further comments. I think the author did a good job on the revision and I do appreciate the authors' effort in the finite-time analysis. In light of this, I will raise my score to 7.

---

> > > ### Author Response · Authors · 2022-08-08
> > > **Thank You!**
> > >
> > > Thank you very much for your feedback, positive evaluation, and support.
> > >
> > > Best,
> > >
> > > Authors.

---

> ### Author Response · Authors · 2022-08-02
> **Authors' Response: Part I**
>
> Thank you very much for the supportive and valuable comments. We address the reviewer's concerns in a point-by-point manner below. Major changes in the revised manuscript are highlighted in blue.
>
> **A.** **Asymptotic convergence results are less informative than finite-time bounds:** Let us address this major concern using two separate arguments.
>
> - First, the condition (P.2)  in Theorem 2.1 --- after reducing the summation bound from $\infty$ to $T$ --- can directly lead to a finite-time complexity bound. Hence, non-asymptotic complexity results are also included implicitly in our framework as a special case. Indeed, since condition (P.2) is typically a consequence of the underlying algorithmic recursion, this is not fully surprising. Let us take SGD as an example to illustrate our reasoning.  Equation (6) --- after taking total expectation --- is the standard recursion for SGD. Unrolling it from $k=0$ to $K=T$ gives $\sum_{k=0}^T \alpha_k \mathbb{E} [||\nabla f(x^k)||^2] \leq M + \mathcal O(\sum_{k=0}^T \alpha_k^2)$ for some constant $M>0$ which then yields $\min_{0\leq k\leq T} \mathbb{E} [||\nabla f(x^k)||^2] \leq (M+\mathcal O(\sum_{k=0}^T \alpha_k^2))/\sum_{k=0}^T \alpha_k$. Choosing the step sizes properly results in the standard finite-time complexity result for SGD. Therefore, finite-time complexity results can be obtained as a byproduct from our analysis. Theorem 2.1 then allows to consider the limiting case $T \to \infty$ and to achieve the (often more challenging) transition from finite-time results to full asymptotic convergence guarantees.
>
> - Second, we argue that asymptotic convergence results can be as informative as finite-time complexity bounds. To support our statement, we have added a new section (Appendix H) in the revised manuscript.  In particular, we constructed a specific example such that the finite-time complexity bound holds, while the gradient norm will never converge to $0$. Such a pathological phenomenon can occur since the finite-time complexity result conveys little information about the last iterate --- especially in nonconvex and stochastic optimization. In this sense, the asymptotic convergence allows to reinforce the finite-time complexity rate: since {$\mathbb{E}[||\nabla f(x^k)||]$}$_{k\geq 0}$ converges to zero, the term $\mathbb{E}[||\nabla f(x^{T})||]$ will stay small (below any predefined threshold) for large $T$. This suggests that, in this situation, it is somewhat safer to return the last iterate of SGD as final output (which is the most common strategy in practice!) and to apply the rates derived from finite-time analysis for it. Therefore,  we believe that both asymptotic and non-asymptotic results are important and informative. Obtaining both convergence guarantees is probably the best option allowing to paint a more complete picture of the convergence behavior of stochastic optimization methods. This is precisely the main motivation of our work: we provide a general convergence theorem that allows to derive (expected and almost sure) asymptotic convergence through a plugin-type analysis, i.e., by verifying the conditions in Theorem 2.1.
>
> At the end of Appendix H, we also state a remark from the blog post of Francesco Orabona (see [31], here [31] refers to the reference [31] in the revised manuscript), which underlines the equal importance of iteration complexity and asymptotic convergence. Parts of the remark are as follows:
>
>  > ''*[...] The common motivation to ignore these past results is that the finite-time analysis is superior to the asymptotic one, but this is clearly false (ask a statistician!). It should be instead clear to anyone that both analyses have pros and cons.*''

---

### Official Review · Reviewer_Nn1D · 2022-07-11

**Rating:** 6
**Confidence:** 3
**Soundness:** 2 fair
**Presentation:** 2 fair
**Contribution:** 2 fair

**Summary:**

This paper derives general convergence results for stochastic optimization methods. The results include the cases where the cost function is nonconvex and nonsmooth.

**Questions:**

I have the following questions about writing as well as the nature of the results:

1) Please give an example, before providing your main theorem in Sec. 2, what an abstract convergence measure \Phi could be.

2) In convex and nonconvex optimization, asymptotic results are somewhat of secondary importance, as nonasymptotic bounds could provide reasonable heuristics to optimize the parameters of optimization methods. Why do the authors provide only asymptotic results in this paper? In other words, what was the difficulty in getting nonasymptotic results?

3) Related to my second question, if authors manage to convert some of these results into nonasymptotic ones, is it clear that the rates of convergence would match standard rates? In other words, someone can derive a very slow convergence rate for these optimization methods (which would all converge in the same way asymptotically), but would not be very useful otherwise.

I would be inclined to reconsider my rating if the authors provide a convincing discussion whether their technique can lead to nonasymptotic bounds.

**Limitations:**

Not applicable.

**Strengths And Weaknesses:**

Strength: The approach is general and can be applicable to several settings.

Weakness: The results are only asymptotic and about the gradient norm (as such, not "global" optimization results).

---

> ### Author Response · Authors · 2022-08-02
> **Authors' Response: Part II**
>
> **B.** **Examples of $\Phi$ before Theorem 2.1 (Q1).** We provide examples of $\Phi$ in the beginning of Subsection 2.1 immediately after the statement of the theorem. The reason why we did not put those exemplary choices of $\Phi$ before Theorem 2.1 is as follows. Theorem 2.1 is not tailored towards showing convergence of specific optimization algorithms. Instead, we present abstract conditions about several abstract sequences based on which we can derive convergence of {$\Phi(x^k)$}$_{k\geq 0}$. Our goal is to make this theorem as general as possible as it may have other applications, e.g., in statistics, other than establishing convergence of optimization algorithms. However, we understand the reviewer's position and we can definitely move these examples before Theorem 2.1 if needed.
>
> We hope that our response and revisions are satisfactory to the reviewer and that all concerns have been addressed appropriately.

---

> ### Author Response · Authors · 2022-08-02
> **Authors' Response: Part I**
>
> Thank you very much for the valuable comments. We address the reviewer's concerns in a point-by-point manner below. Major changes in the revised manuscript are highlighted in blue.
>
> **A.** **Asymptotic convergence versus iteration complexity (Q2 & Q3).** The major concern of the reviewer is about asymptotic convergence versus iteration complexity. In the following, let us address this concern clearly and in detail.
>
> - The condition (P.2) in Theorem 2.1 --- after reducing the summation bound from $\infty$ to $T$ --- can directly lead to a finite-time complexity bound, which should always match the standard ones. Hence, non-asymptotic complexity results are also included implicitly in our framework as a special case. Indeed, since condition (P.2) is typically a consequence of the underlying algorithmic recursion, this is not fully surprising (see the discussions in Phase I of Subsection 2.1). Let us take SGD as an example to illustrate our reasoning.  Equation (6) --- after taking total expectation --- is the standard recursion for SGD. Unrolling it from $k=0$ to $K=T$ gives $\sum_{k=0}^T \alpha_k \mathbb{E} [||\nabla f(x^k)||^2] \leq M + \mathcal O(\sum_{k=0}^T \alpha_k^2)$ for some constant $M>0$. This then yields $\min_{0\leq k\leq T} \mathbb{E} [||\nabla f(x^k)||^2] \leq (M+\mathcal O(\sum_{k=0}^T \alpha_k^2))/\sum_{k=0}^T \alpha_k$. Choosing the step sizes properly results in the standard finite-time complexity result for SGD. Such a quick argument also applies to RR (see its recursion equation (21)), prox-SGD (see its recursion in Lemma 3.4 and Remark 3.7), and SMM (see its recursion in Lemma 3.8). Therefore, finite-time complexity results can be obtained as a byproduct from our analysis. Theorem 2.1 then allows to consider the limiting case $T \to \infty$ and to achieve the (often more challenging) transition from finite-time results to full asymptotic convergence guarantees. We plan to include this discussion and the outlined connection to complexity results in the finalized version of the main paper.
>
> - Asymptotic convergence can be as important as non-asymptotic bounds. Studying the asymptotic convergence properties of stochastic optimization methods is an important direction and has a long history; see the many related references in our manuscript. To further motivate our study, we have added a new section (Appendix H) in the revised manuscript.  In Appendix H, we argue that non-asymptotic complexity and asymptotic convergence are both informative and are complementary to each other. Let us summarize some of the core arguments here. In Appendix H, we constructed a specific example such that the finite-time complexity bound holds, while the gradient norm will never converge to $0$. Such a pathological phenomenon can occur since the finite-time complexity result conveys little to no information about the last iterate --- especially in nonconvex and stochastic optimization. In this sense, the asymptotic convergence allows to reinforce the finite-time complexity rate: since {$\mathbb{E}[||\nabla f(x^k)||]$}$_{k\geq 0}$ converges to zero, the term $\mathbb{E}[||\nabla f(x^{T})||]$ will stay small (below any predefined threshold) for large $T$. This suggests that, in this situation, it is somewhat safer to return the last iterate of SGD as final output (which is the most common strategy in practice!) and to apply the rates derived from finite-time analysis for it. Therefore,  we believe that both asymptotic and non-asymptotic results are important and have advantages and disadvantages. Obtaining both convergence guarantees is probably the best option allowing to paint a more complete picture of the convergence behavior of stochastic optimization methods. This is precisely the main motivation of our work: we provide a general convergence theorem that allows to derive (expected and almost sure) asymptotic convergence through a plugin-type analysis, i.e., by verifying the conditions in Theorem 2.1.
>
> At the end of Appendix H, we also state a remark from the blog post of Francesco Orabona (see [31], here [31] refers to the reference [31] in the revised manuscript), which underlines the equal importance of iteration complexity and asymptotic convergence. Parts of the remark are as follows:
>
> > *''[...] The common motivation to ignore these past results is that the finite-time analysis is superior to the asymptotic one, but this is clearly false (ask a statistician!). It should be instead clear to anyone that both analyses have pros and cons.''*

---

> ### Author Response · Authors · 2022-08-08
> **Feedback & Additional Remarks and Questions (If Any)**
>
> Dear Reviewer Nn1D,
>
> We hope that our response addresses most of your concerns. Specifically, we showed that our current framework can automatically lead to expected iteration complexity results. We also added a new section (Appendix H in the revised manuscript) to show --- by means of constructing a concrete example and citing existing remarks --- that asymptotic convergence can be as important as the iteration complexity bound, thus motivating our major study.
>
> Since the deadline of the discussion period is approaching, we would highly appreciate to receive feedback from you. Please let us know whether our responses are satisfactory. If you have any further questions or remarks, we will be more than happy to provide additional clarifications and details.
>
> Best,
> Authors.

---

> ### Comment · Reviewer_Nn1D · 2022-08-08
> **Thank you**
>
> I thank authors for their insightful response. I have reconsidered my evaluation in the light of their response.

---

> > ### Author Response · Authors · 2022-08-09
> > **Thank You!**
> >
> > Thanks so much for your feedback, positive re-evaluation, and support.
> >
> > Best,
> >
> > Authors.

---

### Official Review · Reviewer_1j9a · 2022-07-12

**Rating:** 7
**Confidence:** 4
**Soundness:** 4 excellent
**Presentation:** 4 excellent
**Contribution:** 3 good

**Summary:**

This manuscript provides a unified asymptotic convergence analysis framework for several centralized stochastic optimization methods such as SGD, random reshuffling, proximal SGD and stochastic model-based methods. By introducing two sets of general conditions on the abstract convergence measure as well as the sequence generated by stochastic optimization algorithms, they derive both the expected and almost-sure convergence to the stationary points, respectively. The authors then apply this result to the abovementioned algorithms to obtain asymptotic convergence guarantees in the expected and almost-sure senses, which either recover existing convergence results (possibly under weaker assumptions) or generate new results.

**Questions:**

- On the generality of the proposed framework: To show the generality of the proposed unified theorem, the reviewer would like to see if it can be applied to other recently developed stochastic optimization algorithms that employs variance-reduction techniques and varying sizes of mini-batches.
- On significance of contribution: It is not very clear how this proposed framework will improve the existing analysis. The reviewer would thus suggests adding a table to compare the convergence results of the existing stochastic algorithms and that obtained from the proposed framework with respect to different assumptions and problem settings to illustrate generality and reveal some insights if any.


**Limitations:**

The generality of the proposed framework is not very clear to the reviewer. The authors are thus suggested to make clear the scope of stochastic optimization algorithms that can be incorporated into this unified convergence analysis.

**Strengths And Weaknesses:**

Originality: The main novelty of this paper relies on the generality of the proposed unified convergence analysis based on the abstract stationarity measure \Phi, which seems new to the reviewer. The major comments are summarized as follows:

Strengths:
- By introducing general conditions on the abstract stationarity measure \Phi, their theoretical analysis enjoys high flexibility to recover or complete the expected and  almost-sure asymptotic convergence results of existing stochastic optimization methods under several problem settings, thus is potential to simplify algorithm design and analysis.

- This paper is well written and easy to follow. The convergence analysis for SGD, RR, prox-SGD and SMM methods is clear and reveals some insights among these methods.

Weaknesses:
- On technical novelty: Since the considered stochastic optimization methods in this work have been well studied, and the proof techniques used in this paper are standard, thus the technical contribution is limited to some extent.
- On the obtained rate results: For non-convex problems, the convergence to a stationary point of the objective function is not very informative since the result cannot declare any optimality, from this understanding, the obtained asymptotic convergence results in this work are even weaker than the existing iteration complexity results which further shows the  non-asymptotic convergence rates under certain measures.

---

> ### Author Response · Authors · 2022-08-02
> **Authors' Response: Part II**
>
> **C.** **Generality of the proposed framework and limitations.** To demonstrate the broad applicability and flexibility of Theorem 2.1, we have provided additional convergence results for SMM and the stochastic proximal point method for solving (13) when $f$ is sufficiently smooth. We also extended the results in Corollary 3.10 by replacing (D.3) with a more general Lipschitz continuity assumption on $f$ and a more common ABC-type variance condition; see Remark 3.11 and Appendix F.3 in the revised manuscript. These results are in line with the convergence results obtained for prox-SGD and, to the best of our knowledge, are new.
>
> In addition, we realized that statement (i) of Theorem 2.1 can be further generalized and that the previous requirement ''$\mu_k \to 0$'' can be weakened to ''{$\mu_k$}$_{k\geq 0}$ is bounded''. This does not require any significant adjustments of the proof of Theorem 2.1. As a result, item (i) of Theorem 2.1 is now also applicable to algorithmic schemes that do not employ diminishing step size strategies.
>
> For algorithms with constant step size that utilize variance reduction techniques or increasing batch sizes, the convergence statements shown in Theorem 2.1 can follow more directly. For instance, following the analysis of prox-SVRG in equation (20) in the supplementary materials of [33] (by Reddi, Sra, Poczos, and Smola), we have $\sum_{s = 0}^\infty \sum_{t=0}^{m-1}[\frac{1}{2\eta}-L]\mathbb{E}[||F_{\mathrm{nat}}^\eta(x_t^{s+1})||^2] < \infty$. Here, $F_{\mathrm{nat}}^\eta$ is the natural residual defined as in our section 3.3.1, $\eta \in(0,\frac{1}{2L})$ is a suitable constant step size, and $s$ and $t$ denote the respective outer and inner iteration numbers of prox-SVRG. Since $\eta$ is fixed, this immediately implies $\mathbb{E}[||F_{\mathrm{nat}}^\eta(x_t^{s+1})||] \to 0$. (This will also follow from Theorem 2.1 (i) with the adjusted condition on {$\mu_k$}$_{k\geq 0}$, but requires some more steps). Therefore, the major application areas of our unified convergence framework are on stochastic optimization methods that have non-vanishing stochastic errors or that utilize diminishing step sizes.
>
> **D.** **Significance of contribution.** We have added a new table (Table 1) in Appendix G in the revised manuscript. Table 1 summarizes and compares available convergence results of several representative and related works. The central value of our work is to provide a simple, plugin-type, and unified way for establishing convergence for a broad class of stochastic optimization methods through verifying the algorithmic properties listed in Theorem 2.1. (As argued in section 2.1, these algorithmic properties can typically build on existing complexity bounds and analyses which is a mild prerequisite). This is in sharp contrast to existing results --- especially for prox-SGD and SMM --- that rely on much stronger and partly non-verifiable assumptions (e.g., surely bounded iterates, almost surely bounded variance, density / Sard-type conditions, etc.) and that utilize much more complicated differential inclusion analysis techniques.
>
> To put our contributions into perspective, let us cite two related paragraphs from the reference [33] (Page 2; after the second informal theorem) on the convergence of prox-SGD:
>
> > *''[...] Consider the case where [the minibatch] is a constant [...], typically the choice used in practice. In this case, the above convergence result no longer holds and it is not clear if prox-SGD even converges to a stationary point at all! To clarify, a decreasing step size [...] trivially ensures convergence [...], but the limiting point is not necessarily stationary. [...]''*
>
> > *''This dismal news does not apply to the convex setting, where prox-SGD is known to converge (in expectation) to an optimal solution using constant minibatch sizes [...]. Furthermore, this problem does not afflict smooth nonconvex problems [...], where convergence with constant minibatches is known [...]. Thus, there is a fundamental gap in our understanding of stochastic methods for nonsmooth nonconvex problems. Given the ubiquity of nonconvex models in machine learning, bridging this gap is important. ''*
>
> We believe that our work can help to address and illuminate some of the questions raised in [33] (i.e., in many applications, prox-SGD with diminishing step sizes will converge to a stationary point).
>
> We hope that our response and revisions are satisfactory to the reviewer and that all concerns have been addressed appropriately.

---

> ### Author Response · Authors · 2022-08-02
> **Authors' Response: Part I**
>
>
> Thank you very much for the supportive and valuable comments. We address the reviewer's concerns in a point-by-point manner below. Major changes in the revised manuscript are highlighted in blue.
>
> **A.** **Technical novelty.** Many of the developed techniques in the proof of Theorem 2.1 are nontrivial compared to the existing analyses, though our overall proof idea is inspired by the convergence analysis of deterministic trust-region methods [8, Theorem 6.4.6] (here, [8] refers to the reference [8] in the revised manuscript). For instance, in the proof of item (i), the main challenge is to deal with the more intricate interactions between different terms and orders (i.e., $a$, $b$, $q$, etc), which is absent in the existing analyses. In the proof of item (ii), we decompose the update into a martingale term $A_k$ and a bounded error term $B_k$. Utilizing this decomposition and results from martingale convergence theory, we can then identify an event $\mathcal M$ that provably occurs with probability $1$ and on which we can conduct sample-based arguments to obtain almost sure convergence. This type of strategy seems to be new and allows to establish almost sure convergence results in a more streamlined and simpler way. Although these technical developments might appear more straightforward or standard, the overall composition and conceptual framework of our proof and theorem is novel, nontrivial, and general (which is underlined by the new convergence results for prox-SGD and SMM obtained in section 3.3 and 3.4 in the paper).
>
> We also consider the proof and convergence guarantees for prox-SGD as one of the (maybe more surprising) technical highlights of the paper. Our results for prox-SGD are based on the more general ABC condition rather than bounded variance assumptions. Here, the key observation is to realize that the additional Lipschitz continuity of $\varphi$ allows to establish a strong link between the function values ''$f(x^k) - \bar f$'' and ''$\operatorname{env}_{\theta \psi}(x^k) - \bar \psi$''. We can then apply Theorem 2.1 to show that prox-SGD converges in expectation and almost surely. Specifically, accumulation points of the stochastic process {$x^k$}$_k$ generated by prox-SGD correspond to stationary points of the problem almost surely and in an expectation sense. Our analysis and unified framework provides the first convergence guarantee for prox-SGD (for nonconvex problems) under common and general assumptions and without imposing restrictive boundedness conditions on {$x^k$}$_k$. Note that Lipschitz continuity of $\varphi$ is typically not stringent as $\varphi$ often acts as a nonsmooth regularizer or as an indicator function; see our Remark 3.3. (See also our later response in **D.** for additional remarks on prox-SGD).
>
> **B.** **Obtained rate results.** Convergence to a stationary point is perhaps the only thing we can expect in nonconvex optimization if no more assumptions on twice differentiability and on the geometry of saddle points and local minima are available.
>
> Furthermore, we have added a new section --- Appendix H --- in the revised manuscript to show that our asymptotic convergence result and the iteration complexity result are equally important --- especially for nonconvex and stochastic optimization. In practice, obtaining both results is probably the best option, which allows to paint a more complete picture of the convergence behavior of stochastic optimization methods. We also refer to our response to *Reviewer qySK* for more discussions along this line.

---

> ### Author Response · Authors · 2022-08-09
> **Additional Remarks and Questions (If Any)**
>
> Dear Reviewer 1j9a,
>
> We hope that our response addresses most of your concerns, questions, and the mentioned weaknesses. In particular, we clarified the technical novelties in Theorem 2.1 and in the convergence analysis of prox-SGD. We have also added a new section in the revised manuscript (see Appendix H) to illustrate that asymptotic convergence can be as important as iteration complexity bounds, thus motivating our major study. Furthermore, we have provided extended convergence results for SMM and stochastic proximal point methods (see Appendix F.3) that can underline and demonstrate generality of our proposed convergence framework.
>
> Since the deadline of the open discussion period is approaching, please let us know whether our responses are satisfactory or if you have any further questions or comments. We will be more than happy to provide additional explanations and details.
>
> Best,
>
> Authors.

---

> > ### Comment · Reviewer_1j9a · 2022-08-09
> > **Thanks for the detailed response**
> >
> > Thank you for carefully preparing the detailed response which have addressed most of the reviewer's concerns and have clarified the scope of the proposed unified convergence theorem (It is also suggested to make it clear the application scope of the proposed unified theorem in the revised manuscript). The reviewer would thus like to raise the score.

---

> > > ### Author Response · Authors · 2022-08-09
> > > **Thank you!**
> > >
> > > Thank you very much for your feedback, positive evaluation, and support! We will include a more detailed and clarifying discussion of the overall application scope of the unified convergence theorem in the main part of the updated manuscript. We believe that such a discussion will be very helpful and we plan to incorporate it in the immediate paragraphs following the presentation of Theorem 2.1. Furthermore, we will put Table 1 (Appendix G) near Subsection 3.5 in the main part of the updated version if the space is enough.
> > >
> > > Best,
> > >
> > > Authors.

---

### Author Response · Authors · 2022-08-08
**Further Questions and Remarks (if any)**

Dear Reviewers,

We hope that our responses address most of your concerns. If you have any further questions or remarks, please let us know and we will be more than happy to answer them before the end of the discussion period.

Best,

Authors.

---

### Meta-Review · Area_Chair_svJG · 2022-08-25

**Recommendation:** Accept
**Confidence:** Certain

**Metareview:**

The authors provide a blanket convergence analysis for several stochastic optimization methods. The techniques are interesting and will be useful.

The authors may want to be a bit more careful on the details on some of their convergence results when they make comparisons. For instance, the main difference between [3] and [26] is in the noise assumptions in [26], which allow to use more aggressive step-size policies. Otherwise, the difference in assumptions that the paper alludes to is reflected in the fact that [26] is getting a stronger convergence result (to a component of critical points), whereas [3] leaves open the possibility that the process escapes to infinity (the assumptions in [26] rule out this behavior). The authors also miss the recent work, which provide a tighter, general characterization:

Y.-P. Hsieh, P. Mertikopoulos, and V. Cevher. The limits of min-max optimization algorithms: Convergence to spurious non-critical sets. In ICML '21: Proceedings of the 38th International Conference on Machine Learning, 2021.


**Award:**

No

---

### Decision · Program_Chairs · 2022-09-14

Accept